# Analysis of Secondary Organic Aerosol Simulation Bias in the Community Earth System Model (CESM2.1)

Yaman Liu[1,2], Xinyi Dong[1,2], Minghuai Wang[1,2], Louisa K. Emmons[3], Yawen Liu[1,2], Yuan Liang[1,2], Xiao Li[1,2], Manish Shrivastava[4]

[1]School of Atmospheric Science, Nanjing University, Nanjing, China
[2]Joint International Research Laboratory of Atmospheric and Earth System Sciences & Institute for Climate and Global Change Research, Nanjing University, China
[3]National Center for Atmospheric Research, Boulder, CO, USA
[4]Pacific Northwest National Laboratory, Richland, Washington, USA

*Correspondence to*: Xinyi Dong (dongxy@nju.edu.cn)

**Abstract.** Organic aerosol (OA) has been considered as one of the most important uncertainties in climate modeling due to the complexity in presenting its chemical production and depletion mechanisms. To better understand the capability of climate models and probe into the associated uncertainties in simulating OA, we evaluate the Community Earth System Model version 2.1 (CESM2.1) configured with the Community Atmosphere Model version 6 (CAM6) with comprehensive tropospheric and

stratospheric chemistry representation (CAM6-Chem), through a long-term simulation (1988–2019) with observations collected from multiple datasets in the United States. We find that CESM generally reproduces the inter-annual variation and seasonal cycle of OA mass concentration at surface layer with correlation of 0.40 as compared to ground observations, and systematically overestimates (69 %) in summer and underestimates (-19 %) in winter. Through a series of sensitivity simulations, we reveal that modeling bias is primarily related to the dominant fraction of monoterpene-formed secondary

organic aerosol (SOA), and a strong positive correlation of 0.67 is found between monoterpene emission and modeling bias in eastern US during summer. In terms of vertical profile, the model prominently underestimates OA and monoterpene concentrations by 37–99 % and 82–99 % respectively in the upper air (>500 m) as validated against aircraft observations. Our study suggests that the current Volatility Basis Set (VBS) scheme applied in CESM might be parameterized with too high monoterpene SOA yields which subsequently result in strong SOA production near emission source area. We also find that

the model has difficulty in reproducing the decreasing trend of surface OA in southeast US, probably because of employing pure gas VBS to represent isoprene SOA which is in reality mainly formed through multiphase chemistry, thus the influence of aerosol acidity and sulfate particle change on isoprene SOA formation has not been fully considered in the model. This study reveals the urgent need to improve the SOA modeling in climate models.

## 1 Introduction

As one of the most important contributors (20 %–90 %) to total fine atmospheric particles (Kanakidou et al., 2004), organic aerosol (OA) plays an important role in the climate system by affecting the radiation budget (Ghan et al., 2012). OA consists

of primary organic aerosol (POA, also called primary organic matter POM) emitted directly from biomass burning, fossil fuels combustion, biological compounds, etc., and secondary organic aerosol (SOA) formed via oxidation of volatile organic compounds (VOCs) (Hallquist et al., 2009; Tsigaridis et al., 2014; Shrivastava et al., 2017). Chamber studies have revealed

the important role of biogenic VOCs such as monoterpenes (Docherty and Ziemann, 2003; Kristensen et al., 2016; Signorell and Bertram, 2009) and isoprene (Kroll et al., 2005; Nguyen et al., 2010; Nguyen et al., 2014; Paulot et al., 2009) in SOA production, and the contribution of anthropogenic VOCs such as aromatic compounds, emitted from vehicle emissions and solvents, also has important influence in urban areas (Nakao et al., 2012; Sato et al., 2010).

Understanding the atmospheric burden and spatiotemporal distributions of OA is one of the key priorities in atmospheric

research because of the central roles it played in regulating both climate and air quality. The radiative forcing effect of OA has been assessed with climate models through tremendous efforts during the past decades (Ghan et al., 2012; Myhre et al., 2013; Sporre et al., 2020; Chen and Gettelman, 2016), yet the limited capability of climate models in terms of simulating the productions and depletions of OA induce large uncertainties. Substantial divergences were reported for models employed in the framework of Aerosol Comparisons between Observations and Models (AeroCom) phase II project even with the same set

of emissions input. The burden of OA varied greatly for 28 models in the range of $0.6 \sim 3.8$ Tg, with OA lifetime ranging from 3.8 to 9.6 days (Tsigaridis et al., 2014). As OA loading and properties of aerosols varied, the estimated radiative forcing of OA ranged from -0.06 to -0.01 W/m$^2$ among the 16 participating models (Myhre et al., 2013), revealing the fundamental uncertainty of OA simulation.

Modeling discrepancies largely come from the lack of a consensus in the representation of chemical composition and formation

processes of OA among different models(Tsigaridis et al., 2014; Goldstein and Galbally, 2007). Although laboratory and chamber studies have revealed thousands of new reactions and new species related to VOCs and SOA, these reactions and species are usually simplified and grouped into a few functions and lumped to fewer species in the models to make it possible for simulating. Many unclear SOA formation processes have to be approximated as the knowledge is still under development (Kanakidou et al., 2004; Hallquist et al., 2009). Thus, different models may use different simplified functions, lumped species

definitions, and approximation methods to represent the overall SOA related processes. For example, SOA chemistry was represented with the two-product method (Lack et al., 2004; Heald et al., 2008) in the earlier model (Lamarque et al., 2012). Since the late 2000s, Volatility Basis Set (VBS) methods (Donahue et al., 2006; Robinson et al., 2007) have been widely adopted by different models due to the advantages over two-products method for considering the volatility (Lack et al., 2004; Heald et al., 2008). Since the 2010s, pilot studies started to include reactive uptake of isoprene epoxydiols (IEPOX) formation

of SOA through aqueous-phase reactions into regional (Shrivastava et al., 2019; Karambelas et al., 2014) or global models (Marais et al., 2016; Zheng et al., 2020; Jo et al., 2019). Despite the tremendous efforts in the early stage, models still underpredicted measured SOA mass concentration by 1-2 orders of magnitude. Hodzic et al. (2016) therefore suggested corrected stronger yields of SOA formation which took into account the influence of vapor wall losses in chamber studies and were considered in Tilmes et al. (2019). However, due to the different gas-phase chemistry, dry and wet deposition schemes,

and heterogenous chemistry schemes, the simulated OA may be different for the same VBS configuration in different models

(Hodzic et al., 2016; Tilmes et al., 2019). These attempts not only validated the parameterizations and chemical pathways derived from measurement studies, but also extended the understanding of SOA formation on a scale broader than the chamber. Once formed, most of OA undergoes chemical aging (Zhang et al., 2007) with volatility and hygroscopicity changing, but such processing is poorly understood due to inadequate relevant observations. While some models consider species-dependent aging reactions and the subsequent volatility change, some simply apply a constant aging rate (Tsigaridis et al., 2014; Jo et al., 2013; Donahue et al., 2006; Zhao et al., 2016). In the remote areas of the United States (U.S.), OA and organic carbon (OC) concentrations show opposite bias in half of participating models (Tsigaridis et al., 2014), indicating the models lack a consensus representation of SOA production and depletion.

To reveal the uncertainties associated with OA simulation in climate models, we evaluate a recent version of Community Earth System Model version 2.1 (CESM2.1) in this study with multiple observational datasets in the U.S. The model has been widely applied for OA climate effect assessment purpose (A. Gettelman et al., 2019; Glotfelty et al., 2017; Tilmes et al., 2019; Jo et al., 2021) and a significant portion of improvements have been implemented in the latest version regarding the chemical mechanisms (Tilmes et al., 2019). In the previous CESM version (Lamarque et al., 2012), SOA chemistry was represented with the two-product method (Lack et al., 2004; Heald et al., 2008). The next big update was reported by Tilmes et al. (2019), in which the two-product method was replaced by VBS following the work by Hodzic et al. (2016). Although CAM-chem has been applied in many studies including the AeroCom program (Tsigaridis et al., 2014), the evaluation of simulated OA concentration hasn't been well documented or thoroughly discussed. We focus on the validation over the U.S. because it has long-term surface measurements and flight campaigns that provide solid observation data. We first evaluate the spatiotemporal characteristics of the simulation bias, and then probe into the chemical mechanism to identify the origins through a series of sensitivity runs, and finally demonstrate the urgent need to both improve the current parameterization of the SOA production scheme and implement a more comprehensive production mechanism in climate model.

## 2 Methods

### 2.1 Model

Community Earth System Model is a coupled Earth System model composed of atmosphere, ocean, land, sea-ice, land-ice, river, and wave models (Danabasoglu et al., 2020). CESM2 (versions 2.0 and 2.1) includes 2 versions of model top, the Whole Atmosphere Community Climate Model version 6 (WACCM6) with 72 vertical layers up to about 150 km and the Community Atmosphere Model version 6 (CAM6) with 32 vertical layers up to about 40 km. CAM6 has simplified chemistry and simplified OA scheme, while CAM6 with comprehensive chemistry and comprehensive OA scheme are called CAM6-Chem which is updated compared to previous versions. The Model for Ozone and Related chemical Tracers (MOZART) chemical mechanism covering the troposphere and stratosphere (referred to as MOZART-TS1) is used in CAM6-Chem. Emmons et al. (2020) reported the updates of MOZART-TS1 in CESM2.1, including the oxidation of isoprene and terpenes, organic nitrate speciation, and aromatic speciation and oxidation, and thus improved representation of ozone ($O_3$) and SOA precursors. The

most recently released CESM2.2 includes a new version (TS2) of MOZART tropospheric chemical mechanism with updates for isoprene and terpene chemistry (Schwantes et al., 2020) aiming at further improving $O_3$ simulation. In this study we also

briefly compare the results between TS1 and TS2 as will be discussed in Sect. 3.3.

Both biogenic and anthropogenic VOCs are considered in CAM6-Chem with improved gas-phase chemical mechanisms (Emmons et al., 2020) and new SOA representation (Tilmes et al., 2019). CAM6-Chem applies the Volatility Basis Set (VBS) scheme (Bergström et al., 2012; Hodzic et al., 2016; Shrivastava et al., 2011; Shrivastava et al., 2013; Donahue et al., 2006; Robinson et al., 2007) by lumping SOA precursors based on their volatility bins to simulate SOA production. In CAM6-Chem,

SOA and the gas-phase condensable sources (SOAG) are categorized into five bins with the saturation concentration (C*) of 0.01, 0.1, 1.0, 10.0, and 100.0 μg/m³ (Tilmes et al., 2019), respectively. Compared with the simple SOA scheme which proportionally calculated SOAG based on emissions of precursors (Liu et al., 2012), this VBS scheme was demonstrated to improve CESM performance with smaller bias of OA concentration over remote regions when evaluated against aircraft observations (Shrivastava et al., 2015; Tilmes et al., 2019). VBS approach relies on empirical parameterizations fitting to

chamber experiments thus the parameters vary between models. The current CAM6-Chem configures VBS scheme and parameters by following the work of Hodzic et al. (2016) with GEOS-Chem as the host model which differs significantly from CAM-Chem in terms of SOA-related modules such as gas-phase chemistry, aerosol dynamics, dry and wet depositions. Consequently , the same scheme and parameterization employed by Hodzic et al. (2016)  may result in different performance within CAM-Chem. Tilmes et al. (2019) provided a comprehensive comparison between simple SOA scheme and the VBS

scheme in terms of simulated SOA burden and radiative forcing, and the simulations were validated against two flight campaigns, yet the evaluation against ground surface measurements at different temporal scale (e.g., annual, seasonal) over different geophysical areas hasn't been thoroughly discussed.

## 2.2 Observations

Surface measurements from the Interagency Monitoring of Protected Visual Environments (IMPROVE) network and multiple

aircraft campaigns in northern hemisphere are incorporated in this study to validate model performance and also facilitate the analysis of SOA trend. IMPROVE is a long-existing program currently managed by the U.S. Environmental Protection Agency (EPA), and is designed to measure chemical composition of ambient fine particles and its spatial and temporal information (Solomon et al., 2014). We used 1988–2019 daily data for seasonal cycle and long-term trend evaluation. As IMPROVE measures organic carbon (OC) mass instead of OA, the concentration of OA observation is derived with the ratio of OA to OC

(OA/OC) which is determined by the aging process. IMPROVE data has been widely employed in many studies for model evaluation purpose (Hodzic et al., 2016; Shrivastava et al., 2015; Tsigaridis et al., 2014). The OA/OC of IMPROVE data varies between sites and the mean value used in this study is 1.8 (1.79 ~ 2.02) as recommended by Malm and Hand (2007). It should be noticed that not all sites have observations for the whole study period, and thus only those (140 sites, locations shown in Fig.1) with more than 10-years continuous data are used in this study to avoid measurement bias.

Free troposphere measurements from a total of five aircraft field campaigns are employed to validate simulated vertical OA profiles including CalNex (California Nexus, Ryerson et al. (2013)), DC3 (Deep Convective Clouds and Chemistry, Barth et al. (2015)), SENEX (Southeast Nexus, (Warneke et al., 2016)), SEAC4RS (Studies of Emissions and Atmospheric Composition, Clouds and Climate Coupling by Regional Surveys, (Toon et al., 2016)), FRAPPE (Front Range Air Pollution and Photochemistry Éxperiment (Flocke et al., 2020). Flights were located in the area of contiguous United States (CONUS)

and took place between 2010 and 2014 with more details presented in Table 1 and Fig. 3.

## 2.3 Simulation configurations

To reproduce the observed meteorological conditions and allow direct comparison with OA measurements, a specified dynamic simulation (SD) is conducted from 1987 to 2019 with a spin-up time of one year thus our discussion can focus on chemical mechanism performance of CAM6-Chem. This experiment (referred as CAM-Chem-SD) uses FCSD component set

in which CAM6 physics, troposphere/stratosphere chemistry (MOZART-TS1) with VBS SOA scheme, historical emission, and offline meteorological field are applied. In details, Temperature, horizontal winds, and surface fluxes are nudged to Modern-Era Retrospective analysis for Research and Applications (MERRA2) fields for detailed comparisons to field experiments and specific observations. The horizontal resolution is $0.9° \times 1.25°$ and vertical resolution is 32 levels with model top at ~45 km (Emmons et al., 2020). Prescribed historical sea surface temperatures (SSTs) are used in the FCSD component

set. Anthropogenic and biomass burning emissions from 1987 to 2014 are from the standard Coupled Model Intercomparison Project round 6 (CMIP6) (Eyring et al., 2016) simulations, and emissions after 2014 are from SSP585 scenario which is based on the shared socioeconomic pathway 5 (SSP5) (O'Neill et al., 2017) and forcing levels of Representative Concentration Pathways 8.5 (RCP8.5). Biogenic emissions are calculated with the Model of Emissions of Gases and Aerosol from Nature (MEGAN) in CESM (Guenther et al., 2012; Emmons et al., 2020). Moreover, another simulation of CESM2.2 which is the

latest released version of CESM, referred to CAM-Chem-SD(TS2), is conducted with MOZART-TS2 gas phase chemistry (Schwantes et al., 2020) from January 2013 to February 2014 with first 2 months as spin-up time. The FCSD component set is also used in CAM-Chem-SD(TS2). Except for the difference of gas phase chemistry, the SOA scheme is also improved in CAM-Chem-SD(TS2) compared with CAM-Chem-SD. The NOx dependence of SOA formation in CAM-Chem-SD(TS2) is not considered in CAM-Chem-SD. Thus, we compared CAM-Chem-SD and CAM-Chem-SD(TS2) to investigate the impact

of NOx dependence on SOA formation.

The current VBS scheme in CAM6-Chem represents SOAG production from nine precursors through 15 reactions. The 15 reactions and chemical formulas of related species from Emmons et al. (2020) are shown at Table S1 and S2. To identify their contributions to total SOA mass and associated simulation uncertainties, we conducted 15 sensitivity experiments with one of the reactions turned off in each experiment. The 15 sensitivity simulations and a 14-month baseline simulation are set up from

January 2010 to evaluate the contribution of each reaction to total SOA production in VBS scheme. To exclude the influence of potential extreme meteorology condition or emission inputs, these sensitivity runs are configured with FC2010climo component set and Newtonian relaxation time of three hours. The FC2010climo component set is as same as FCSD component

set except that the emissions are a 10-year average used for each year of the simulation. As one reaction is turned off for each sensitivity experiment, the difference between base and the sensitivity experiment represents the influence of this specific reaction (Table 2). For instance, the no ISOP-OH-SOAG experiment excludes SOAG produced by ISOP + OH reaction (reaction 1 in Table S2) with other configurations same as the base run (CAM-Chem-climo) experiment.

## 3 Results and Discussion

### 3.1 Evaluation against IMPROVE

We first validate the surface OA simulation with IMIPROVE data for the CAM-Chem-SD scenario and find an overall significant overestimation, with the results and statistics presented in Fig. 1 and Table 3. As demonstrated in Fig. 1, CESM2.1 shows substantial simulation bias with normalized mean bias (NMB) varying from -73.87 % to 176.47 % at different sites with daily data pairs against IMPROVE measurements. The model noticeably overestimates annual average surface OA concentration over continental U.S. (CONUS) with mean bias (MB) of 0.41 μg/m³ and NMB of 20.27 %. It also shows large regional difference as demonstrated by the sharp contrast between the two subdomains, eastern US (EUS, 65°–95° W, 25°– 49° N) and western US (WUS, 95°–125° W, 25°–49° N) as presented in Fig. 1. Prominent overestimation is found over EUS with NMB more than 100 % at nine sites, while moderate underestimation is shown over WUS. Our validation suggests that the updated SOA representation (by applying VBS) helps improve the performance of CAM6-Chem as compared with its earlier versions. CAM4-Chem was reported to underestimate OC by ~36 % compared with IMPROVE dataset with correlation coefficient of 0.41 (Lamarque et al., 2012), and CAM5-Chem was found to overestimate with NMB by 24 % at urban sites and 217 % at remote sites (Tsigaridis et al., 2014). The AeroCom phase II multi-model ensemble mean showed NMB of -48 % (range -85 % to 24 %) at urban locations and -70 % (range -38 % to 217 %) at remote locations on global scale, suggesting that the performance of CESM2.1 is consistent with other climate models.

Annual and seasonal variations of surface OA from the model and IMPROVE are presented in Fig. 2 to examine the model's capability of reproducing the temporal variation. The model generally reproduces the annual decreasing trend of OA concentration for the whole CONUS domain, and is in good agreement with observation in spring and fall, but substantially overestimates in summer and slightly underestimates in winter as shown in Fig. 2(a). Monthly average of the simulation is found to have larger overestimation in warmer months (May–Sep.) than cooler months as shown in Fig. 2(b). In EUS, the simulation prominently overestimates surface OA concentration in summer by 4.26 μg/m³ (131.15 %) but successfully reproduces the temporal change with a strong correlation with observations of 0.60 (Table 3) as shown in Fig. 2(c). As compared with CONUS domain, surface OA concentration from the simulation at EUS shows an even greater overestimation during warmer months as shown in Fig. 2(d). In WUS, simulated OA shows a slow decreasing trend in summer (-0.02 μg/m³ per decade) while the observations indicate an ascending trend (0.23 μg/m³ per decade). The large inter-annual variation from 1999 to 2019 are shown in observed surface OA concentration mainly due to the influence of wildfires (Malm et al., 2017). The simulated surface OA concentration also has large inter-annual variation but do not shows increasing trend due to lower

value after 2017. It needs to be emphasized that historical emissions are used from 1987 to 2014 and SSP585 emissions after 2014 in CAM-Chem-SD simulation, which means the emissions do not exactly match the observed condition after 2014. The model shows smaller bias in WUS but also a poor correlation of 0.36 (Table 3) in summer as shown in Fig. 2(e) and 2(f). The large modeling bias of OA in summer over EUS may be attributed to the discrepancy in simulating SOA, because POA is proportionally determined by emissions from fossil fuel and biofuel that have smaller seasonal diversity (see supplementary

material Fig. S1). Biomass burning also contributes POA, but wildfire or prescribed fire emission is minimal in EUS (van der Werf et al., 2017). Production of SOA is represented by the MOZART-TS1 and VBS module as mentioned in Sect. 2. MOZART-TS1 has been demonstrated in a few studies (Emmons et al., 2020) to overestimate summertime surface $O_3$ over southeast US, thus it may subsequently induce uncertainties while simulating the VOCs chemistry. Besides, the performance and uncertainty of the VBS scheme in CAM6-Chem remain largely undocumented regarding whether the VBS scheme tends

to overestimate or underestimate. With the same VBS scheme but implemented in GEOS-Chem, Hodzic et al. (2016) reported simulation bias of 34.8 % through validation against IMPROVE data, and the bias was explained by evaporation of OC from IMPROVE samples (Kim et al., 2015) and uncertainties in the boundary layer parameterization. The performance of CAM-Chem-SD seems better as our simulation bias for CONUS domain is 20.27 %. But with the larger biases in simulated OA spatial and temporal trends shown in Fig. 1 and Fig. 2, it is critical to realize that the overall evaluation statistic is averaged

from large overestimations and underestimations from sites across the CONUS domain.

## 3.2 Evaluation against flight campaigns

We find systematic underestimation in the upper air through the model evaluation against Aerosol Mass Spectrometry (AMS) measurements from flight campaigns, with the profiles shown in Fig. 3 and evaluation statistics summarized in Table 1. These campaigns were conducted over land area of North America during 2010–2014 and most of the measurements were collected

in summer with rest of the data in late spring and early fall. Simulations are paired with measurements by choosing the nearest model grid at the corresponding time at hourly scale. OA concentrations are underestimated in upper air (>500 m) by -20 % – -70 % as compared with the aircraft measurements. The model shows larger bias when compared with SENEX, DC3 and SEAC4RS that mostly cover EUS, while the vertical profiles measured in WUS (CalNex and SENEX) are reproduced better with stronger correlation coefficient and smaller bias. The most prominent bias is found in the comparison with DC3 campaign

at ~7 km (geometric altitude above mean sea level) due to the aircraft samples of smoke plume from the High Park fire west of Ft. Collins (Barth et al., 2015), suggesting that fire emission plume structure is not properly represented because of the coarse model grid resolution. In general, the model is found to notably underestimate upper air OA for all flight campaigns with larger discrepancy over EUS. We also examine the simulated and observed isoprene and monoterpene profiles during the SEAC4RS campaign, as shown in Fig. 3(g). These two biogenic VOCs (BVOCs) are reproduced well between surface and ~2

km altitude, but are also substantially underestimated at >2 km height, suggesting too fast oxidation of these VOCs at high altitudes or uncertainties in parameterizations of convective transport and vertical mixing. Validation against the recent ATom (Atmospheric TOMography aircraft campaign) measurements from Wofsy et al. (2018) also suggested underestimation in the

upper air over marine areas (see supplementary material Fig. S2 and Table S3), consistent with the recent multi-model evaluation results reported in Hodzic et al. (2020). The photolytic depletion of SOA is represented in VBS scheme of CAM6-

Chem by following Hodzic et al. (2016) which demonstrated this stronger removal process can help lower the GEOS-Chem model overestimation of upper air OA over remote areas (mostly over ocean). But the underestimations of high-altitude SOA within CAM6-Chem model over CONUS land area are different from GEOS-Chem, and are likely related to model-to-model differences in physics, aerosol chemistry, and wet removal.

### 3.3 Uncertainties in VBS schemes

This section reveals the contributions from different SOA precursors and probes into the uncertainties within the current VBS scheme in CAM6-Chem. We first investigate the seasonality and diagnose the components of simulated OA to narrow down the most important candidates causing the modeling bias. Figure 4(a)–(c) present the CAM-Chem-climo simulated surface OA concentration with the contributions from POA and five bins of SOA (SOA1–SOA5) over CONUS, EUS, and WUS. On annual average scale POA consists 35.6 % of the total OA with larger contribution in winter (~50 %), while SOA dominates

in summer (~77.1 %) over CONUS. Our CAM-Chem-climo simulation suggests that the overall ambient OA concentration is determined by SOA for CONUS domain, which is also consistent with observation-based studies (Yu et al., 2004; Zhang et al., 2018). Figure 4(a) shows that the absolute contribution from POA is relatively less dynamic than SOA through the year although both show higher values in summer, suggesting that the seasonality of surface OA is primarily determined by SOA. The five volatility bins show consistent seasonality with high concentrations in warmer months but the two least volatile bins

SOA1 and SOA2 show more prominent seasonal variations, and these two bins also contribute most to total SOA. The dominant influences of SOA1 and SOA2 on total OA presented by CAM6-Chem is consistent with other VBS-based models (Farina et al., 2010).

We then examine the contribution from each of the 15 VBS reactions (listed in Table S1) to identify the most influential species over CONUS, EUS, and WUS as shown in Fig. 4(d)–(f). The SOA formed by each reaction is calculated as the difference

between CAM-Chem-climo and the corresponding sensitivity experiment described in Table 2. The 16 sensitivity experiments (Table 2) are designed for avoiding nonlinear uncertainty of SOA formation mechanisms. For instance, $SOA_{MTERP+O3}$ is calculated by the difference between SOA from CAM-Chem-climo scenario and SOA from "no MTERP+O3" scenario. In CAM-Chem-climo scenario, the reaction is simulated to calculate monoterpene (MTERP) oxidation by $O_3$ and produce SOAG (Table S1). In "no MTERP+O3" scenario, the reaction is still simulated as usual, but SOAG yield is set to zero to avoid the

potential nonlinear influence of turning off the reaction. The sensitivity simulations demonstrate dominant contributions over CONUS and EUS is from 3 BVOC-formed SOA, lumped monoterpenes (MTERP), followed by isoprene (ISOP) and sesquiterpenes (BCARY), in summer, while the intermediate volatility organic compounds (IVOCs) and aromatics (BENZENE, TOLUENE, and XYLENES) are mainly emitted by anthropogenic sources and play a more important role in winter. The contribution from the 15 reactions over WUS show similar results except that SOA formed by ISOP show less

contribution compared to SOA formed by BCARY due to less $SOA_{ISOP+OH}$ in summer. SOA formed from aromatics also show

much smaller seasonal variation as compared with BVOC-formed SOA because anthropogenic emissions are relatively stable throughout the year, while biogenic emissions are substantially more intensive during the growing season than other periods. Besides, SOA formed by MTERP dominates total SOA vertical distribution (~34.5%) as shown in Fig. S3. The sensitivity experiments demonstrate that monoterpene-formed SOA plays the most important role in determining both surface and vertical simulated OA concentration in summer on average over the CONUS domain.

We further diagnose the SOA concentration in July formed from these reactions because of the prominent seasonality (larger bias in summer) and regional difference (larger bias in EUS) revealed in earlier sections. Figure 4(g)–(i) present the contributions from each pathway to the five SOA bins in July over CONUS, EUS, and WUS. Among the 15 pathways to form SOA, the reaction of MTERP+$O_3$ forms most SOA, followed by MTERP+OH and ISOP+OH over CONUS and EUS, while SOA formed by MTERP+OH contributes most over WUS, which is consistent with the over estimation of $O_3$ over EUS in MOZARY-TS1 as mentioned in Emmons et al. (2020). We also find that MTERP-formed SOA is mostly in the second VBS bin SOA2 with low volatility, but most of SOA form by ISOP+OH is in the fourth VBS bin SOA4 with high volatility.

The contribution of monoterpene-derived SOA is consistent with other measurement studies (Xu et al., 2018; Zhang et al., 2018) which demonstrated that a significant fraction of observed OA was monoterpene generated SOA over southeast US. Zhang et al. (2018) indicated monoterpene derived SOA contributed to ~42 % of total AMS $PM_1$ OA during SOAS (Southern Oxidant and Aerosol Study) field campaign. As Sect. 3.1 reveals substantial overestimation in summer and minor bias in winter, it is thus very likely that monoterpene-related VBS reactions may be primarily responsible for the modeling uncertainty. We have found a strong correlation between monoterpene emission and modeling bias as shown in Fig. 5. The CAM-Chem-SD simulation bias increases along biogenic monoterpene emission with a correlation of 0.41 in summer and -0.12 in winter over the CONUS domain. This relationship is particularly strong (correlation of 0.67) for summer over EUS (Fig. 5(g)) with higher biogenic VOCs, and almost negligible in winter over WUS (correlation of -0.05) with lower biogenic emission. Moreover, we also find biogenic VOCs flux dominates OA bias with higher correlation in summer over EUS, while anthropogenic VOCs flux dominates OA bias over WUS as shown in Fig. S4. Despite the fact that isoprene biogenic emission flux shows the same correlation as monoterpene biogenic emission flux over EUS, the prominent seasonality of modeling bias and the strong correlation with summer time emission suggests that monoterpene SOA dominates the performance of CAM6-Chem over abundant biogenic emission region.

The mass yields of VBS in CAM6-Chem was parameterized following Hodzic et al. (2016) which adjusted the values used in GEOS-Chem (Jo et al., 2013) to account for the wall-loss. These parameters are shown in Table 4. The adjustment by Hodzic et al. (2016) resulted in stronger SOA production and led to overestimation in surface OA concentration compared with AMS global network and biogenic sources contributing to the total SOA production. It is certainly reasonable to take the wall-loss effect into account when making the chamber measurements. But it also should be noticed that those measurements were conducted under artificial environment with predefined chemical species that may vary significantly from the real meteorology condition and atmospheric chemistry regime. Thus, the parameters reported in the chamber studies need to be carefully interpreted and adjusted when applied in atmospheric models. The current VBS scheme in CAM6-Chem differs from Hodzic

et al. (2016) by considering the differences of oxidants but doesn't consider the $NO_x$ dependence of monoterpene oxidation. The mass yields applied in CAM6-Chem are generally in the middle of low and high $NO_x$ conditions used by Hodzic et al. (2016), but still show substantial overestimation due to monoterpene SOA as mentioned above, suggesting that the mass yields in CAM6-Chem require further adjustment.

This simplified representation of VBS parameterization also affects simulation bias because BVOCs have been demonstrated to be closely influenced by $NO_x$ concentration (Shrivastava et al., 2017). Since high $NO_x$ usually suppresses the formation of monoterpene SOA, observed OA would be lower at places with high $NO_x$ concentration than those with low $NO_x$. Subsequently, the overestimation due to monoterpene SOA would be more severe at places with high $NO_x$ level. Fig. 6 (a)–(d) presents the simulation bias at two typical IMPROVE sites representing the high (Agua Tibia: 33.5° N, 117.0° W) and low (Lake Sugema: 40.7° N, 92.0° W) $NO_x$ conditions respectively. With comparable levels of monoterpene emissions, simulation bias at Agua Tibia is substantially higher than that at the Lake Sugema especially during Sep. and Oct. when $NO_x$ concentrations are maximum. We further investigate the relationship between OA bias and $NO_x$ concentration at low MTERP biogenic emission as shown in Fig. 6(e). OA bias increases with higher $NO_x$ concentration at same MTERP biogenic emission in summer at 53 sites. A recent study (Jo et al., 2021, 2020) reported the development of CAM6-Chem for implementing the NOx-dependent yields, along with several other updates including isoprene epoxydiols (IEPOX) derived SOA through heterogeneous chemistry, isoprene emission adjustment, biogenic VOCs deposition, and more detailed gas-phase chemistry of isoprene. We also evaluated simulation from CESM2.2 (CAM6-Chem configured with MOZART-TS2 which includes the above-mentioned updates except for heterogeneous chemistry of IEPOX-SOA) against IMPROVE observations through a full year simulation (2013.03 - 2014.02). We find that these updates moderately lower the simulated OA concentration, but surface OA is still substantially overestimated in summer. The observed OA was 2.4 $\mu g/m^3$ at IMPROVE sites over the CONUS domain in July, 2013, while simulated OA by CESM2.1 and CESM2.2 were 5.1 and 4.2 $\mu g/m^3$, respectively (Fig. S5). Thus, we suggest that the monoterpene SOA yield in current CESM2.1 and CESM2.2 might be configured too high and lead to surface OA overestimation, and the VBS parameterization for other BVOCs in CESM may also need further adjustment for reducing OA bias.

In addition to the monoterpene parameterization, representation of isoprene SOA by pure gas-phase VBS may also induce critical uncertainty to the model simulation, since over certain regions like the southeast USA, isoprene SOA is mostly formed from IEPOX via irreversible multiphase chemistry that is closely affected by gas-aqueous phase transfer and acid catalysed reactive uptake, for which neither has been considered in CESM2.1. A few recent studies have revealed that IEPOX- SOA is influenced by anthropogenic emission because of the changes in inorganic aerosol acidity and sulfate particles (Shrivastava et al., 2019; Zheng et al., 2020). Both modeling and measurement evidence have suggested that the steadily decreasing anthropogenic $NO_x$ and $SO_2$ emission have led to the reduction of biogenic SOA over western and southeastern US during the past decade, yet CAM-Chem-SD has difficulty to reproduce this trend as shown in Fig. 7. The IMPROVE data shows a decreasing trend of 0.527 $\mu g/m^3$ per decade at Cape Romain NWR, but CAM-Chem-SD simulated trend is 0.098 $\mu g/m^3$ per decade. Ridley et al. (2018) indicated the reduction of anthropogenic emissions, due to EPA regulations on vehicular sources

and power generation, can explain more than two-thirds of decline in OA over CONUS from 1990–2012. Marais et al. (2017) showed $SO_2$ emissions controlled biogenic OA concentration by sulfate-BVOCs interaction in summer over EUS.

## 4 Summary and Conclusions

The goal of this study is to understand the performance of CESM2.1 and reveal the remaining biases for the simulation of SOA. Through validation against surface measurements and flight campaigns over the US, we have found that CESM2.1 is able to capture inter-annual and seasonal variation of surface OA concentration with a correlation coefficient by 0.41, but systematically overestimates surface OA concentration in summer by 68.78 %. Larger summer time bias is found over eastern US (overestimates by 131.15 %) where BVOCs emissions are more intensive than western US. Opposite to the overestimation near the surface, consistent underestimations by -20 % – -70 % of OA in the upper air are found in the validation against all five flight campaigns.

Our analysis suggests that it is likely simulated monoterpene SOA production is parameterized with too high yields and may be the most influential factor that affects the modeling bias for three reasons: first, monoterpene SOA contributes most (46.3 %) to the total SOA in summertime, while other anthropogenic POA or BVOCs has a smaller impact on the severe overestimation. The large contribution of monoterpene SOA simulated by CAM6-Chem is consistent with other measurement and modeling studies, but the current VBS configuration adopted from GEOS-Chem may require further adjustment. Isoprene may also play an important role in modeling uncertainty but the influence is likely less significant than monoterpene as the isoprene-derived SOA consists 17.0 % of total SOA in summer. Second, the simulation bias showed a strong spatiotemporal correlation with monoterpene emission as demonstrated by the large overestimation in summer over eastern US, and larger overestimation of OA is found at places with higher $NO_x$ condition under same monoterpene emissions level. Third, overestimation of OA at surface layer and underestimation of OA and monoterpene in the free troposphere suggests that both the production and photolytic removal processes might be parameterized too strong.

It shall be noticed that our analysis of OA simulating bias mainly focused on SOA production, while other related processes such as vertical transport, removal through wet and dry depositions, and evaporation also affect the simulated OA concentration. Model treatments for these processes may also induce uncertainties to the simulation results. The uncertainty related to VBS, however, might play a more important role than other processes according to the evaluation results shown in the above section. For example, CESM has been demonstrated to reproduce well the vertical profiles of meteorology variables (He et al., 2015). A recent sensitivity study (Gaubert et al., 2020) showed the systematic underestimation of CO vertical profile was mainly due to bias in anthropogenic emission other than the vertical transport scheme. For dry and wet depositions, CESM showed slightly higher estimations than other models with the AeroCom II program, but the estimated lifetime was well consistent with the multi-model-mean (Tsigaridis et al., 2014). Evaporation of OA was significantly slower than the rapid reaction rates of VOCs and the oxidants (Vaden et al., 2011; Emmons et al., 2020). Most importantly, these processes are unlikely to dominate the spatiotemporal pattern of OA bias demonstrated in this study, which was found to be highly correlated with biogenic SOA

instead. While uncertainty within these processes is more likely to result in systematic bias, the spatiotemporal pattern of OA bias and results from sensitivity simulations both implied the critical role of the chemical production process, although it is undoubtedly necessary to identify and address the potential uncertainty within other related processes in the future. It is also necessary to note that, the five aircraft campaigns discussed in this study were all conducted in warm months. The simulated

performance in winter and the discrepancy between near and above surface OA bias needs further investigation. The updated VBS scheme under different $NO_x$ condition and multiphase isoprene-derived SOA chemistry have been recently considered in other global and regional models which showed better performance in SOA simulation (Hodzic et al., 2016; Shrivastava et al., 2019; Jo et al., 2021), revealing the urgent need for improving process-level representations of SOA formation and removal in models.


**Code and data availability.** The CESM model code is available at https://www.cesm.ucar.edu/models/cesm2/release_download.html. Observational data for DC3 (https://doi.org/10.5067/Aircraft/DC3/DC8/Aerosol-TraceGas), FRAPPE (https://doi.org/10.5067/Aircraft/DISCOVER-AQ/Aerosol-TraceGas), SEAC4RS (https://doi.org/10.5067/AIRCRAFT/SEAC4RS/AEROSOL-TRACEGAS-CLOUD), and

ATom (https://doi.org/10.3334/ORNLDAAC/1581; Wofsy et al., 2018) can be obtained from the NASA LaRC data archive. CalNex (https://csl.noaa.gov/groups/csl7/measurements/2010calnex/P3/DataDownload/) and SENEX (https://esrl.noaa.gov/csd/groups/csd7/measurements/) data are available via the NOAA ESRL data archive. IMPROVE data are available at http://vista.cira.colostate.edu/Improve/improve-data/.

**Author contributions.** XD and Yaman L conceived the idea and designed the model experiments. Yaman L performed the

simulations, conducted the analysis, and wrote the paper. MW, XD, Yawen L and XL provided conceptualization for the paper. Yuan L, Yawen L and MS provided guidelines for the simulations in CESM v2.1. Yawen L helped in IMPROVE data downloading and processing. XD, Yaman L, LEK, MS and MW edited the paper.

**Competing interests:** The authors declare that they have no conflict of interest.

**Acknowledgements.** This work is supported by the National Natural Science Foundation of China [grant numbers 91744208, 41925023, 41575073, and 41621005] and the Ministry of Science and Technology of the People's Republic of China [grant number 2016YFC0200503]. This research was also supported by the Collaborative Innovation Center of Climate Change,

Jiangsu Province. We greatly thank the High Performance Computing Center of Nanjing University for providing the computational resources used in this work. The CESM project is supported primarily by the United States National Science Foundation (NSF). This material is based upon work supported by the National Center for Atmospheric Research, which is a major facility sponsored by the NSF under Cooperative Agreement No. 1852977. We thank all the scientists, software engineers, and administrators who contributed to the development of CESM2. We acknowledge Pedro Campuzano-Jost and

Jose-Luis Jimenez of the University of Colorado-Boulder for the DC3 AMS aerosol composition data. We acknowledge Anna Middlebrook and Roya Bahreini for the CalNex dataset. We acknowledge the use of observations from the FRAPPE campaign, which was funded by the National Science Foundation (NSF) and the State of Colorado. We acknowledge Joost de Gouw of the University of Colorado-Boulder for help with the SENEX AMS data. We acknowledge NASA for providing the SEAC4RS and ATom AMS data.

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

**Table 1: Aircraft measurements used in this study**

| Campaign | Dates | Region |
|---|---|---|
| CalNex (Ryerson et al., 2013) | 2010.4.30~ 6.22 | California and the eastern Pacific coastal region |
| DC3 (Barth et al., 2015) | 2012.5.1~ 6.30 | northeastern Colorado, west Texas to central Oklahoma, and northern Alabama |
| SENEX (Warneke et al., 2016) | 2013.6.01~ 7.10 | southeast U.S. |
| SEAC4RS (Toon et al., 2016) | 2013.8.6~ 9.24 | southeast and west US |
| FRAPPE (Flocke et al., 2020; Dingle et al., 2016) | 2014.7.26~ 8.19 | northern front range metropolitan area (central U.S.) |

675

**Table 2: CESM experiments used in this study**

| Index | Experiment ID |
|-------|---------------|
| B1 | CAM-Chem-SD |
| B2 | CAM-Chem-SD (TS2) |
| B3 | CAM-Chem-climo |
| E1 | no ISOP+OH |
| E2 | no ISOP+O3 |
| E3 | no ISOP+NO3 |
| E4 | no MTERP+OH |
| E5 | no MTERP+O3 |
| E6 | no MTERP+NO3 |
| E7 | no BCARY+OH |
| E8 | no BCARY+O3 |
| E9 | no BCARY+NO3 |
| E10 | no BENZENE+OH |
| E11 | no TOLUENE+OH |
| E12 | no XYLENES+OH |
| E13 | no IVOC+OH |
| E14 | no SVOC+OH |
| E15 | no GLYOXAL |

**Table 3:** The correlation coefficient (CC), mean bias (MB) and normalized mean bias (NMB) between observations (five field campaigns and IMPROVE surface measurements) and CAM-Chem-SD.

| Observations | | CC | Mean Obs. ($\mu g/m^3$) | Mean Sim. ($\mu g/m^3$) | MB ($\mu g/m^3$) | NMB (%) |
|---|---|---|---|---|---|---|
| **IMPROVE** | | | | | | |
| **CONUS** | Annual | 0.40 | 2.07 | 2.48 | 0.41 | 20.27 |
| | Spring | 0.67 | 1.65 | 1.57 | -0.08 | -4.81 |
| | Summer | 0.37 | 2.90 | 4.87 | 1.97 | 68.78 |
| | Fall | 0.34 | 2.23 | 2.23 | -0.01 | 0.13 |
| | Winter | 0.70 | 1.49 | 1.22 | -0.27 | -19.05 |
| **EUS** | Annual | 0.79 | 2.64 | 3.72 | 1.08 | 40.82 |
| | Spring | 0.64 | 2.49 | 2.66 | 0.17 | 6.71 |
| | Summer | 0.60 | 3.26 | 7.52 | 4.26 | 131.15 |
| | Fall | 0.69 | 2.63 | 2.88 | 0.25 | 9.70 |
| | Winter | 0.82 | 2.25 | 1.82 | -0.43 | -19.11 |
| **WUS** | Annual | 0.36 | 1.78 | 1.89 | 0.11 | 10.49 |
| | Spring | 0.77 | 1.23 | 1.04 | -0.19 | -15.36 |
| | Summer | 0.48 | 2.72 | 3.66 | 0.94 | 34.83 |
| | Fall | 0.35 | 2.03 | 1.91 | -0.12 | -5.85 |
| | Winter | 0.73 | 1.12 | 0.91 | -0.21 | -18.00 |
| **Aircraft** | | | | | | |
| **CalNex** | | 0.43 | 2.06 | 1.46 | -0.60 | -29.01 |
| **DC3** | | 0.12 | 2.99 | 0.72 | -2.17 | -72.75 |
| **SENEX** | | 0.33 | 7.09 | 4.22 | -2.87 | -40.54 |
| **SEAC4RS** | | 0.10 | 6.90 | 1.93 | -4.97 | -71.97 |
| **FRAPPE** | | 0.27 | 3.05 | 2.42 | -0.63 | -20.64 |

680

**Table 4: Mass yield coefficient of monoterpene in Jo et al. (2013), Hodzic et al. (2016) and CAM6-Chem in VBS bins. C* is the saturation concentration ($\mu g/m^3$).**

| Log(C*) | Condition | Jo et al. (2013) Low NOx | | Hodzic et al. (2016) low NOx | high NOx | CAM6-Chem - | |
|---|---|---|---|---|---|---|---|
| | Oxidant | OH, $O_3$ | $NO_3$ | OH, $O_3$, $NO_3$ | | OH, $O_3$ | $NO_3$ |
| ≤-2 | | | | 0.093 | 0.045 | 0.0508 | |
| -1 | | | | 0.211 | 0.015 | 0.1149 | |
| 0 | | 0.01 | 0.07 | 0.064 | 0.142 | 0.0348 | |
| 1 | | 0 | 0.06 | 0.102 | 0.061 | 0.0554 | 0.17493 |
| 2 | | 0.54 | 0.24 | 0.110 | 0.074 | 0.1278 | 0.59019 |
| 3 | | 0 | 0.41 | 0.125 | 0.165 | | |

685

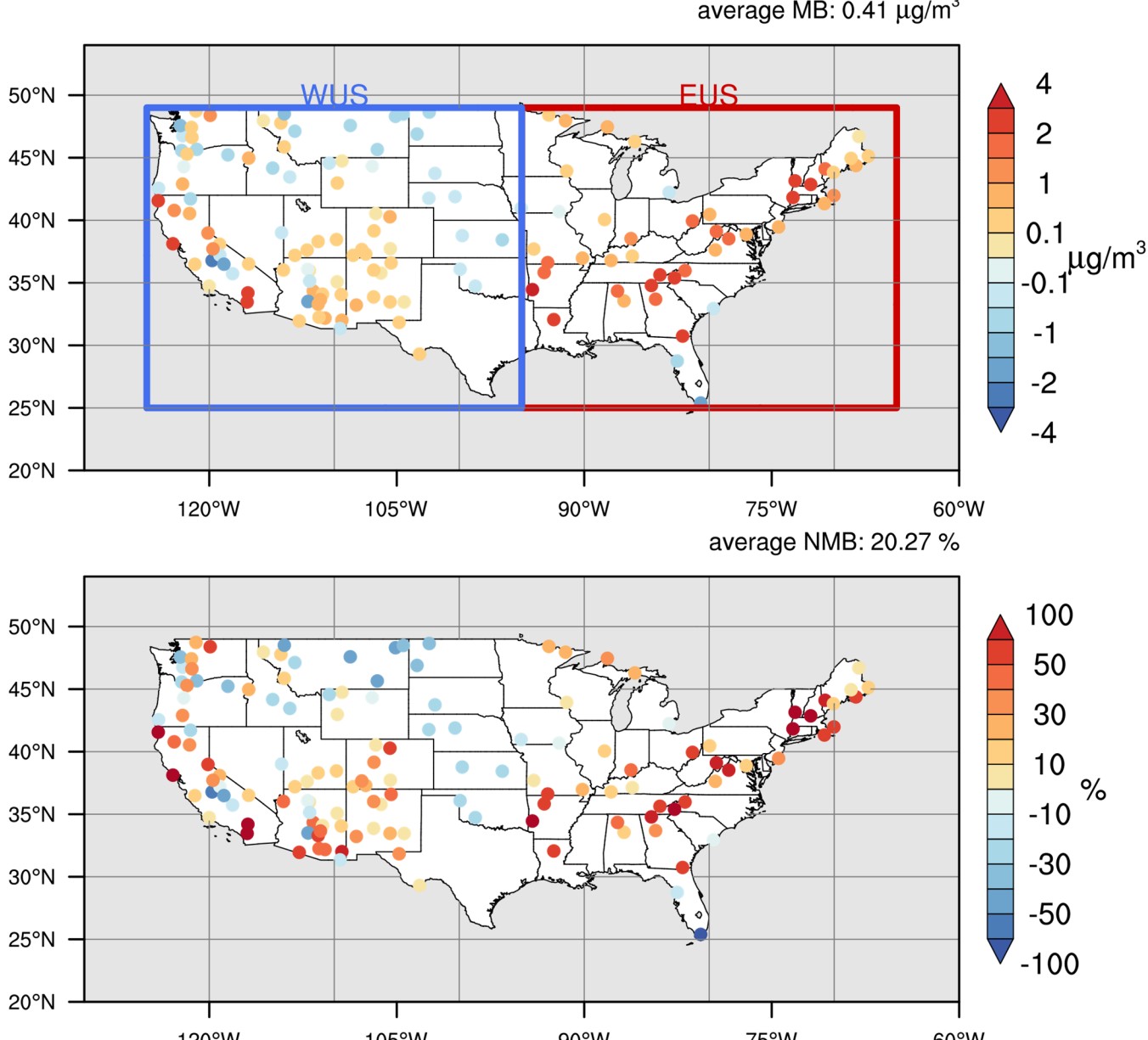

**Figure 1: 1998–2019 CAM-Chem-SD surface OA concentration mean bias (top figure, unit: μg/m³) and normalized mean bias (bottom figure, unit: %) compared to IMPROVE data. CONUS is divided into two subdomains, EUS (red box) and WUS (blue box).**

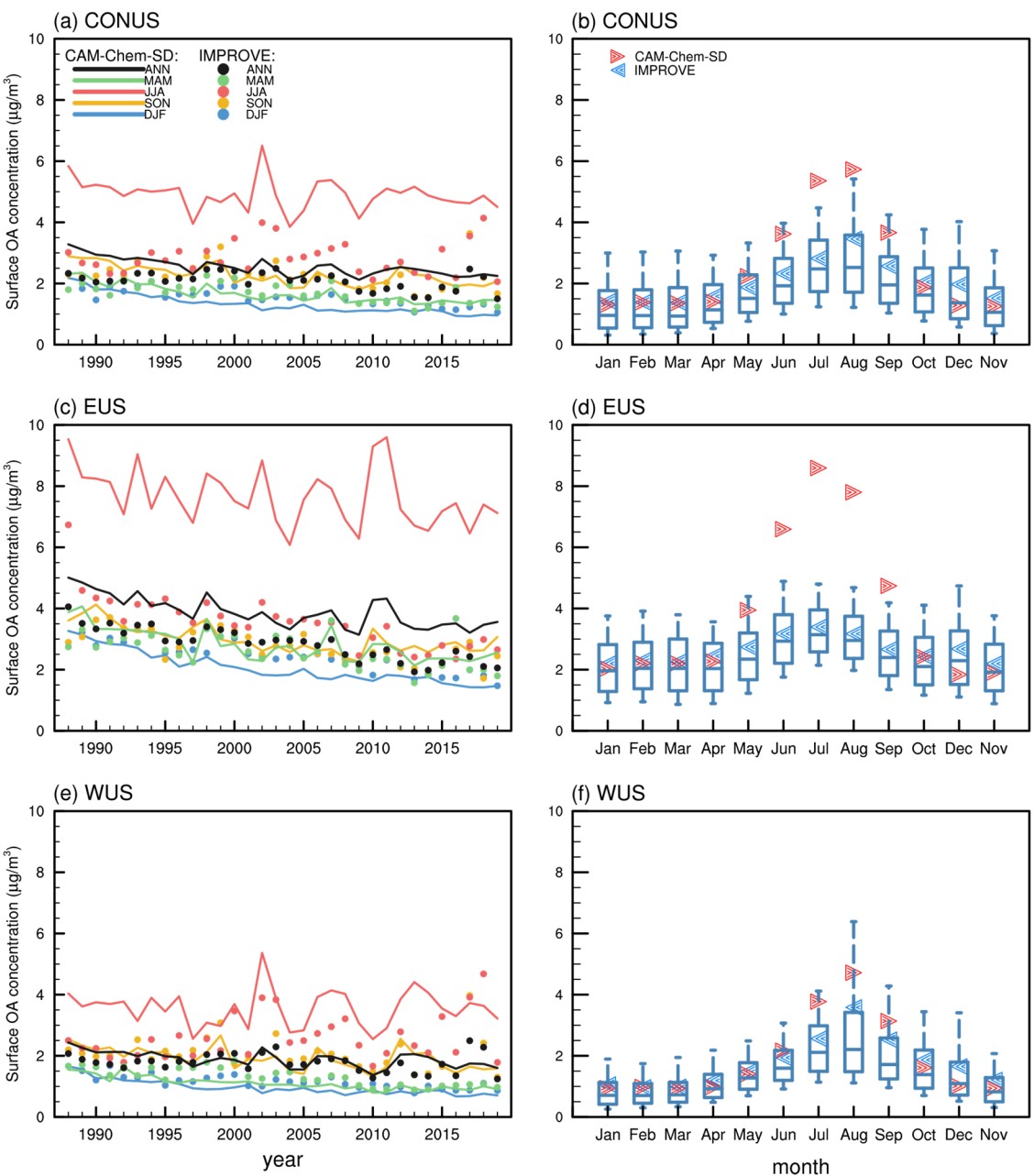

**Figure 2:** The left column is 1988–2019 annual mean (black lines) and seasonal average (blue lines for winter, green lines for spring, red lines for summer, and yellow lines for autumn) surface OA concentration of IMPROVE (solid dots) and CAM-Chem-SD (solid lines) over US continent (a), eastern U.S.(c), and western U.S.(e). The right column is seasonal cycle of 1988–2019 average surface OA concentration of IMPROVE (blue dots) and CAM-Chem-SD (red dots) over CONUS (b), EUS (d) and WUS (f). Every blue box denotes the 10th, the 25th, the median, the 75th and the 90th percentiles of the observations for all selected sites in each month.

695

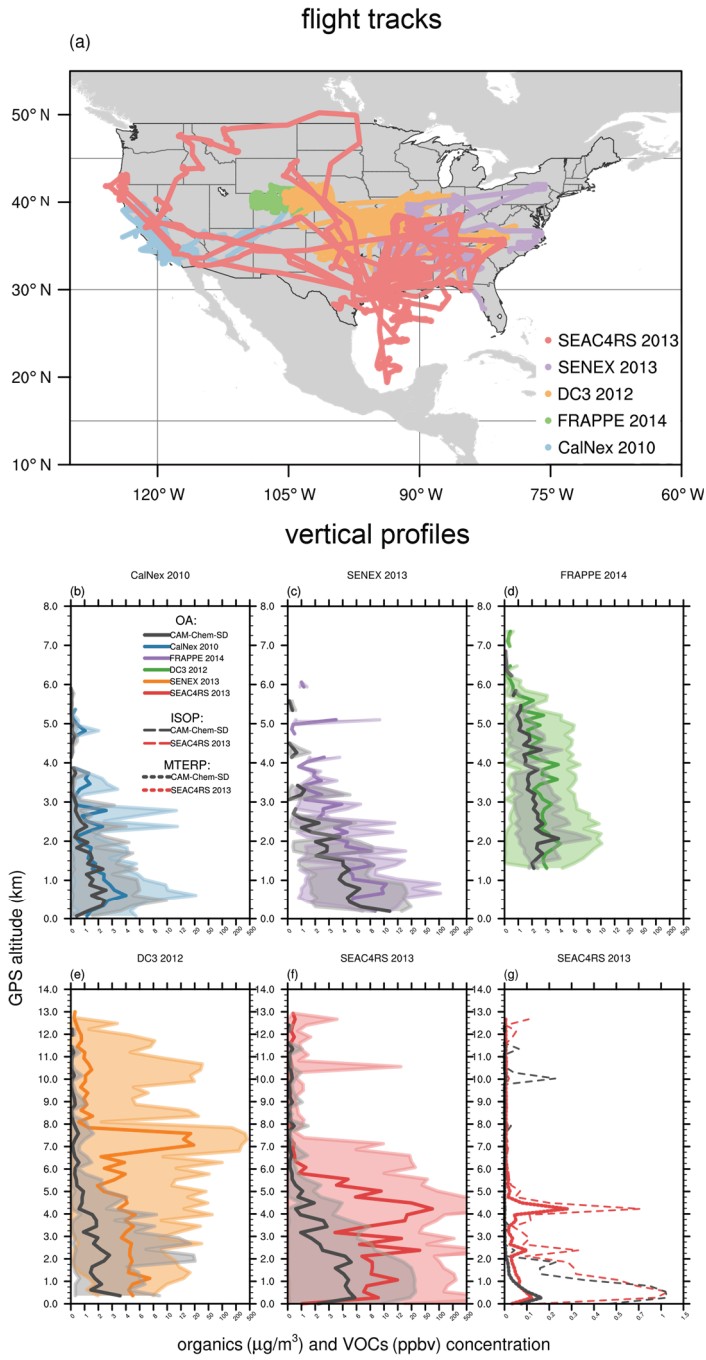

**Figure 3: Flight tracks of five aircraft campaigns, and vertical profiles of average OA (solid lines, panels b-f), ISOP (dashed lines, panel g) and MTERP (dotted lines, panel g) concentration from CAM-Chem-SD (black), CalNex (blue), SENEX (purple), FRAPPE (green), DC3 (orange) and SEAC4RS (red) flight campaigns. The range of OA concentration at each layer is showed as the shaded area or dashed lines.**

700

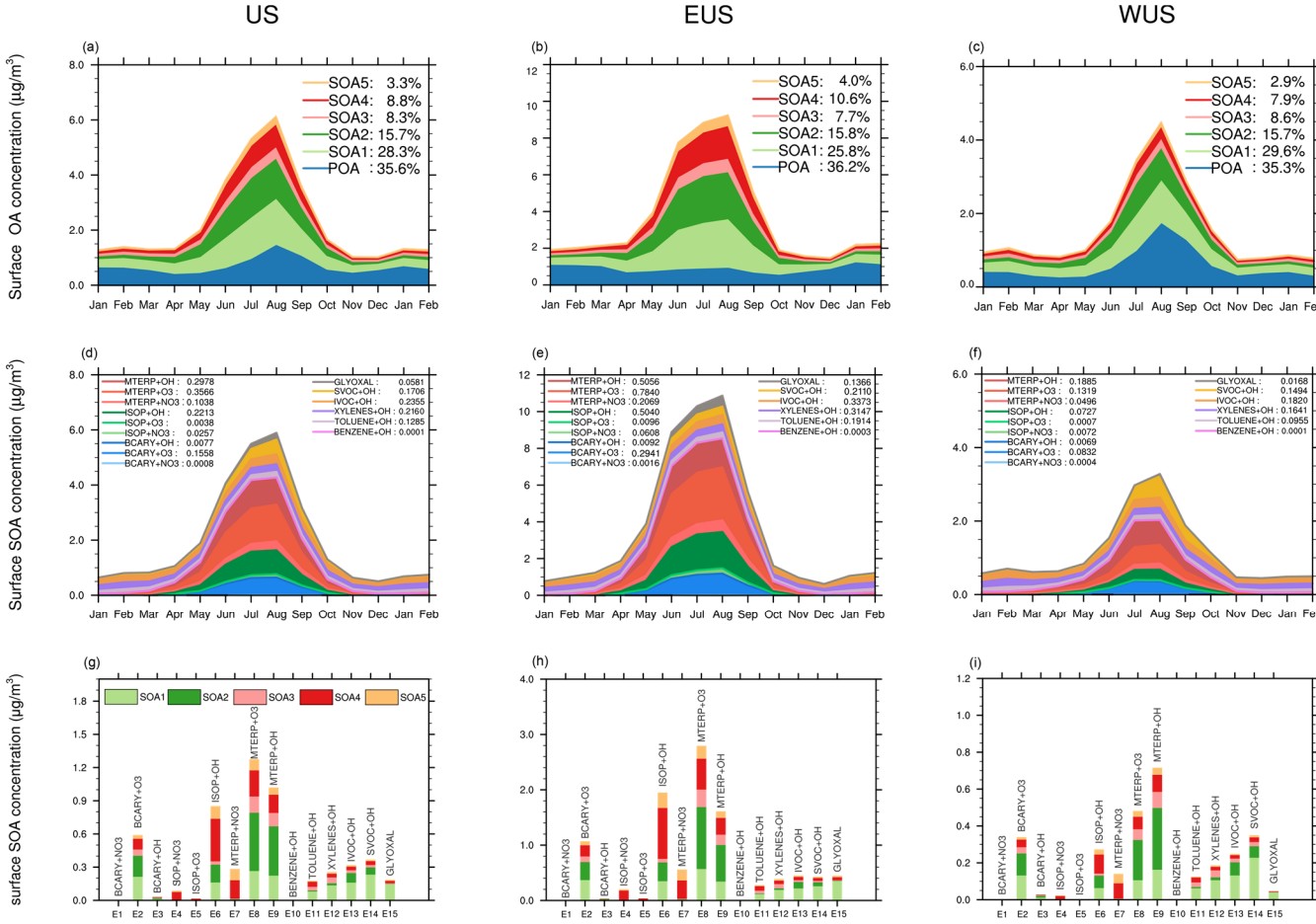

**Figure 4:** (a)~(c) Seasonal variation of surface OA concentration from CAM-Chem-climo simulation at all IMPROVE sites located over (a) CONUS, (b) EUS and (c) WUS. The contribution of POA and SOA in five VBS bins are shown in filled areas. The 14-month average contribution of POA and SOA are shown as the number in the legend. (d)~(f) Seasonal variation of SOA formed by 15 different pathways over (d) CONUS, (e) EUS and (f) WUS. The contribution of 15 reactions are shown in filled areas and 14-month average contribution is shown as the number in the legend. (g)~(i) the contributions from each pathway to the five SOA bins over (g) CONUS, (h) EUS and (i) WUS in 2010 July. SOA concentration in 5 bins are shown in filled bars.

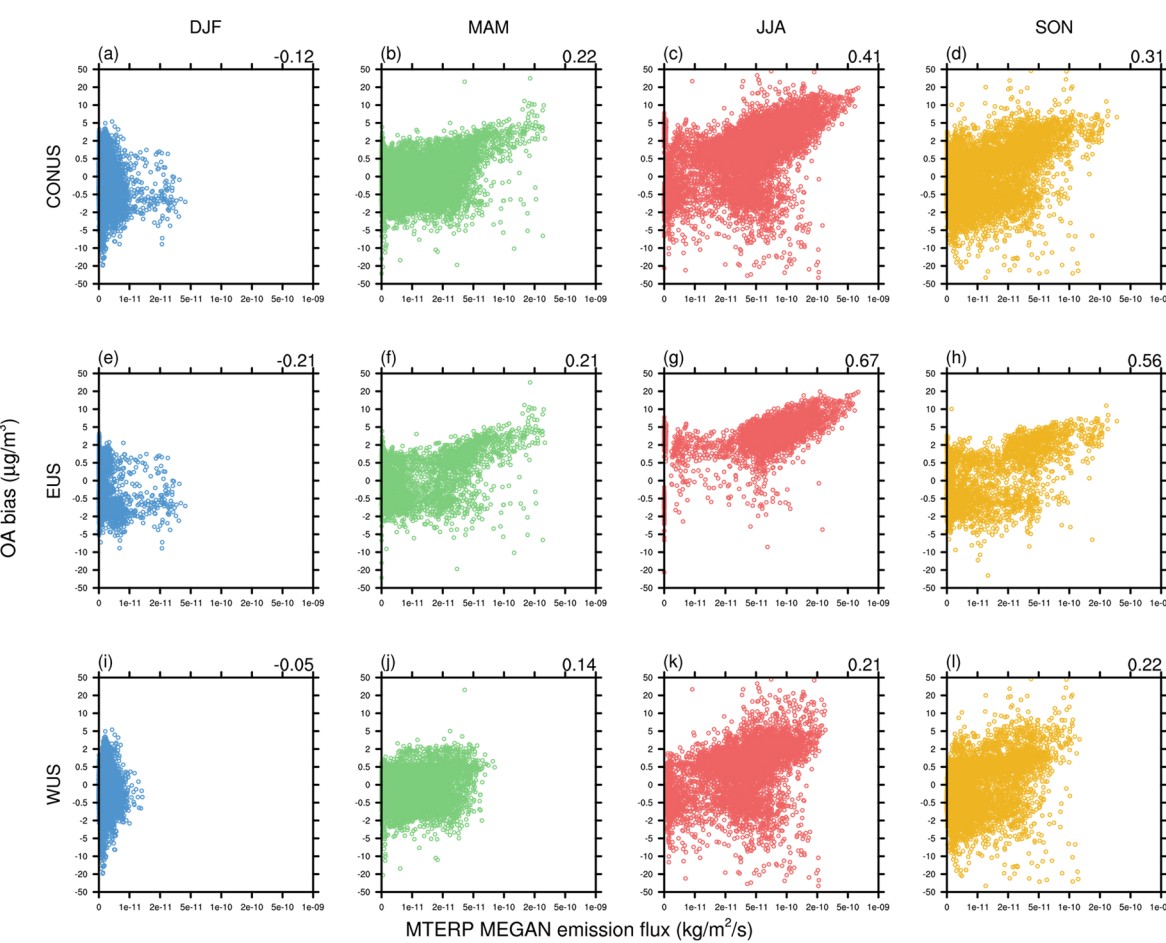

**Figure 5: The relationship between simulated surface OA bias (Y axis) and surface MTERP emissions (X axis) in DJF (a, e, i), MAM (b, f, j), JJA (c, g, k) and SON (d, h, l) over CONUS (a–d), EUS (e–h) and WUS (i–l). The correlation coefficient of OA bias and MTERP emission is shown as the number in the top right of each plot.**

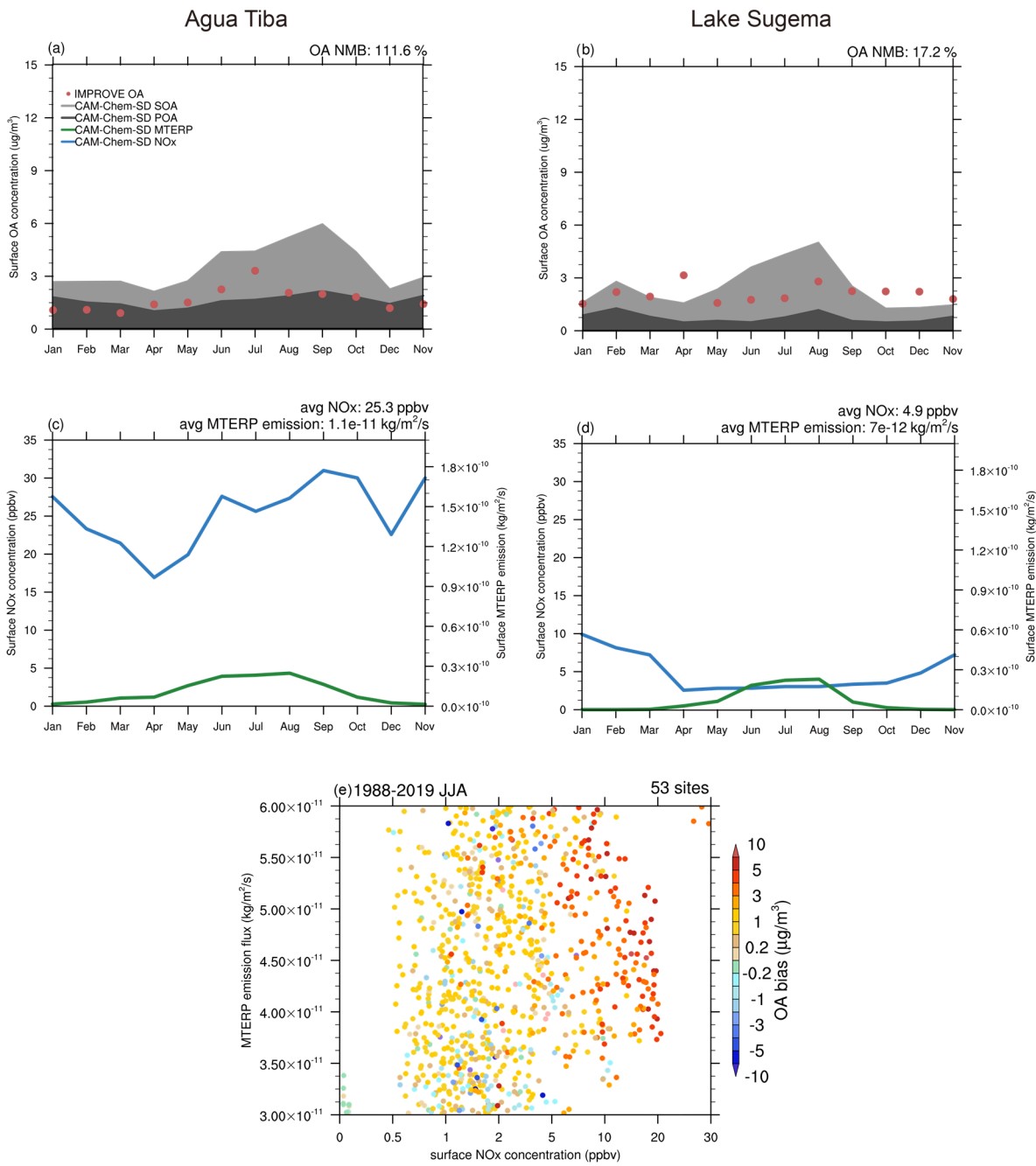

**Figure 6:** 2010 observed surface OA concentration (red dots) and simulated POA (dark filled area) and SOA (gray filled area) at Agua Tibia site (33.5° N, 117.0° W) (a) and Lake Sugema site (40.7° N, 92.0° W) (b); simulated surface $NO_x$ concentration (blue lines) and MTERP emission from MEGAN (green lines) at Agua Tibia site (c) and Lake Sugema site (d). 1988–2019 summertime surface OA bias (colorful dots) at specific $NO_x$ concentration and MTERP emission flux (e) over 53 sites where MTERP emission flux are between $3 \times 10^{-10}$ and $6 \times 10^{-10}$.

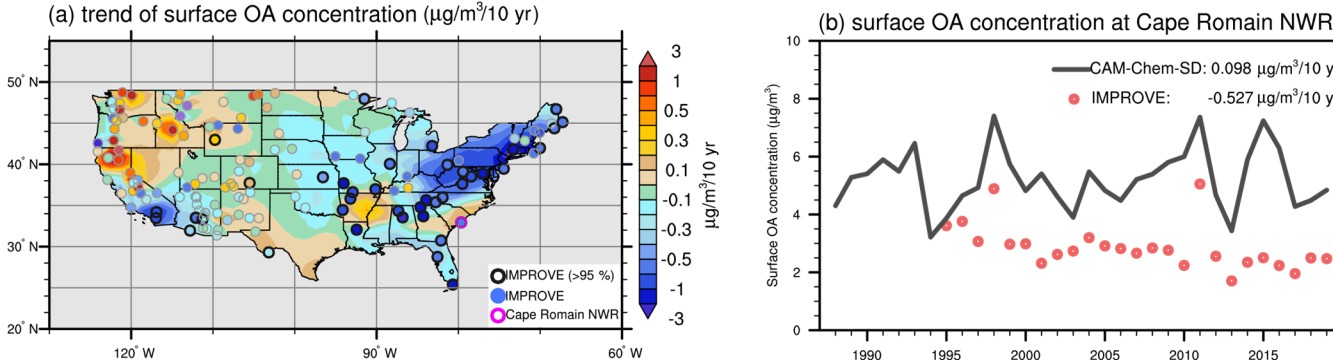

**Figure 7: 1988–2019 JJA (a) simulated surface OA decade trend and IMPROVE surface OA decade trend at selected IMPROVE sites (filled circles). Black circles indicate sites with statistically significant trends with 95 % confidence according to the student's T-test; (b) simulated (black lines) and observed (red dots) surface OA concentration at Cape Romain NWR site (shown as purple circle in Fig. 7(a)). Simulated and observed OA decade trends (unit: μg/m³/10 year) in summer at the site are shown as the numbers in the legend.**