# Peer review of "Analysis of Secondary Organic Aerosol Simulation Bias in the Community Earth System Model (CESM2.1)"

_Atmospheric Chemistry and Physics, 2020_

## Referee Comment (RC1) · Anonymous Referee #1 · 18 Feb 2021

**General comments**

This is an interesting and well written paper. They study evaluates organic aerosol in CESM2.1 over the United States by comparing the model to long term surface measurements and aircraft campaigns. The authors find that the model overestimate organic aerosol during summer. Moreover, the model comparison with flight campaigns reveal that the model underestimate organic aerosol in the upper air. The authors conclude that these results could be explained by too high monoterpene SOA yields which result in too strong SOA production close to monoterpene sources.

The topic of this paper falls well in the scope of ACP. The scientific methods in the paper are sound and well explained. The authors explain and discuss the results in

the figures and tables in an clear and interesting manner. I recommend the paper for publication in ACP after the following comments have been addressed.

**Specific comments**

- The model used in this study, CAM6-Chem differs from the standard CAM6 since it has a more advanced chemistry. My impression is that CAM6 is the standard atmospheric model in CESM2.1. Could you describe to what extent CAM6-Chem is used in comparison to CAM6? It would be beneficial to better clarify the differences between CAM6 and CAM6-Chem in the methods section. In part of the method you describe the changes in CESM2.1. It would be nice to refer to CAM6 or CAM6-Chem instead, as the text is currently written it is difficult to know if the VBS scheme is included in both CAM6 and CAM6-Chem or only in the latter. Moreover, you have evaluated CAM6-Chem, but it would be interesting to know how well CAM6 performs in comparison to CAM6-Chem with respect to organic aerosol. What is the the differences in performance between CAM6-Chem and CAM6 in terms of organic aerosol?

- The model is only compared to observations over the United States. Could you comment on the limitations of this and if the model has been evaluated in other locations in any other studies.

**Technical corrections**

- Line 771,:"over previous versions" sounds a bit odd.

- Line 138-141: This is a very long sentence, please split it up.

- Line 169: "prominently overestimates in" is there a word missing here? What is overestimated?

- Line 170: "with a strong correlation with observations of 0.60 as shown in Fig. 2c" The correlation coefficients are not shown in Fig 2 but rather in Table 3. Please refer to the table or both the figure and table. The same problem with referring to figure 2 instead of table 3 occur on line 174.

- Line 170: "As compared with CONUS domain, simulation at" are there missing words in this part of the sentence?

- Line 295-299. This is a very long sentence that should probably be split up. Also, the English in this sentence needs to be checked.

---

## Referee Comment (RC2) · Anonymous Referee #2 · 25 Feb 2021

Review of Analysis of Secondary Organic Aerosol Simulation Bias in the Community Earth System Model (CESM2.1)

This manuscript presents a comparison between simulated organic aerosol (OA) by the Community Earth System Model (CESM2.1) and measured OA from surface and aircraft observations. The authors demonstrate that simulated OA is over predicted in the summer months, likely due to an overprediction of secondary organic aerosol (SOA). The authors perform a suite of sensitivity simulations, turning off one reaction per simulation in the OA chemical mechanism, and conclude that SOA production through monoterpenes is the likely cause of the simulated OA overestimation in the summer.

[Figure]

The authors also note large OA underpredictions aloft (compared to aircraft observations) and moderate underpredictions at the surface in winter (compared to surface observations). Atmospheric chemical models continue to struggle to accurately represent OA. While there have been a number of previous model/measurement comparison studies on OA, the difficulty in simulating OA warrants additional publications. The subject matter in this study is useful and falls within the scope of ACP. However, I have concerns regarding the extent of the analysis and presentation of findings that prevent me from recommending publication at this time.

General Comments:

1. The argument that the model overestimation of OA at the surface and in summer is caused by an overrepresentation of monoterpene SOA production needs to be better substantiated or the limitations of this assertion better discussed. To be clear, I think the authors make a good suggestion by pointing to the large monoterpene SOA burden and the correlation with model bias. However, the correlation with model bias is not enough to make this argument. For instance, the authors point out that isoprene SOA also has a positive correlation with model bias. I don't completely follow why isoprene SOA was dismissed as a reason (Line 385). Isoprene also appears to have a seasonal cycle that peaks in the summer and the authors note the correlation with isoprene and model bias is also positive. Could the over prediction be due to both monoterpene and isoprene SOA yields? Additionally, the authors state in the Conclusions (L 362) that the other POA and SOA components cannot explain the model bias; however, this does not appear to be explicitly shown in the Results section. I do see that monoterpene SOA dominants the OA composition in the summer (and I agree this is a good candidate for the cause of model overestimation), but I do not see a discussion that the other SOA species could not also contribute to the model overestimation. Again to be clear, I agree that the monoterpene SOA yield is a good suggestion for the cause of the model overestimation, but I feel this argument needs more context. This paper could be improved with a better discussion of this argument and its limitations or an additional

simulation with altered monoterpene yields that reduced the model bias.

2. A large limitation of this study is that it focuses only on SOA production as opposed to any other chemical or physical processes in the model. Could the over estimation of model OA at the surface and underrepresentation of OA aloft point to an issue with vertical transport or removal? What about evaporation of OA - is this included in the model? I believe this paper would benefit from an explicit discussion of this limitation and how it affects the results.

3. The sensitivity simulations turn off chemical reactions one at a time; however, these chemical mechanisms are not necessarily linear (or additive). How does this assumption affect your results?

4. The Introduction Section would benefit from further discussion on SOA oxidation and chemistry (including the VBS scheme) as well as a literature review of previous studies focused on model/measurement comparison of OA. Additionally, a number of statements are lacking citations (see Specific Comments for examples). This topic (of OA representation in models) has been explored previously. As I mentioned at the start of this review, the continued challenge of representing OA in models certainly warrants continued study. That said, I believe this study could be improved by including a literature review of previous measurement/modeling studies in the Introduction. I note that the authors do point to and comment on previous studies in the Methods and Results Sections (which is great). However, I think the manuscript would be improved by clearly discussing relevant previous work in the introduction. This would improve the ability of readers to follow the comparisons in later sections. One such example of this, is that Hodzic et al. (2016), which is cited throughout this work, seems to argue for stronger SOA production rates and stronger SOA sinks. Conversely, this study seems to argue for the opposite. This is an interesting comparison that could use more context.

Specific Comments:

1. I do not think an appropriate color scale was chosen for Figure 1. While red and blue are certainly appropriate for opposite ends of a diverging color scale, it is not immediately obvious to me that green should be opposite yellow. I suggest the authors use a standard diverging color scale (or simply shades of red and blue on each side).

2. The units of the color scale in Figure 1 could be more obvious. I suggest including the units as the color bar label itself or at least in the caption.

3. The legend of Figure 4 is much too small to be readable.

4. The panel labels in the caption of Figure 4 are inconsistent. Sometimes the panel label follows the description while other times it precedes the description.

5. The caption in Figure 7 points to the wrong color (I think it should read "black line" instead of red).

6. The units in Figure 7a are never stated. Please be explicit about units in all figures.

7. Table 3 is difficult to read. Could the different regions be grouped in a more obvious way?

8. Please add units to Table 3.

9. This is a minor comment - the citations need spaces after the semicolon.

10. The sentence at Line 51 is unclear. Are you comparing the model representation to other model processes or stating the reasons why model representation of OA is challenging?

11. Lines 51-57 should be edited or revised for clarity. I am not sure what is meant by these sentences.

12. Citations are needed for the comments on OA and climate impacts (such as on Line 41 and Line 60).

13. Citations are needed for the sentences that begin on Line 49, 50, and 52.

14. Line 80/81 - I think the papers by Donahue et al. (2006) and Robinson et al. (2007) should be included in the citation for the VBS.

15. Line 85 - minor typo with the comma

16. Line 124 - Could you elaborate (briefly) on what a "specified dynamical" simulation is? In addition, please define "FCSD" and "FC2010climo".

17. L 152 - I do not think the word choice of "critical" simulation bias is correct. Do you mean "large" or "substantial"?

18. L 173 - I suggest changing "descending" trend to "decreasing" trend.

19. L 173 - I think this is a really interesting point that needs a little clarification. Do you mean that the increasing wildfires are leading to increasing trends in observations but the wildfire emissions are not included in the model and so the trend is not represented?

20. L 184 - Is the current study not also influenced by the bias of evaporation of OA off filters as in Hodzic et al. (2016)?

21. Line 321 - Is this an entirely new model configuration and simulation? If so, I recommend including this model configuration in the Methods Section. If I understand correctly, this simulation includes SOA production schemes that were suggested as part of the model bias in the previous paragraph. This seems like an important result that should be given more discussion.

22. Line 356-357 is confusing. Is the second parenthetical placed correctly?

23. Line 368 states "...and photolytic removal processes might be too strong". I do not follow why this is an argument in support of monoterpene SOA production being too high. It seems like it argues that bias is not entirely due to SOA production rates in contrast to the point of this paragraph.

---

## Author Comment (AC2) · 9 Mar 2021

Please find our response in the attached file.

Please also note the supplement to this comment:
https://acp.copernicus.org/preprints/acp-2020-1182/acp-2020-1182-AC2-supplement.pdf

---

## Author Response (AR1)

We are very grateful to the evaluations from the reviewers, which have allowed us to clarify and improve the manuscript. Below we addressed the reviewer comments, with the reviewer comments in black and our response in blue.

Before we provide the detailed point to point reply, we provide an overview of main changes and improvements:

1. All the figures and the tables are moved to the end of the manuscript.

2. The reference list is adjusted according to the standards of Copernicus style.

3. We include a literature review of previous measurement and modeling studies in the Introduction section.

4. We add some detailed descriptions of the model and the experiments in Methods section.

5. We modify the legend of Fig. S4 in the supplement.

**Reply for the referee comment#1**

**General comments:** This is an interesting and well written paper. They study evaluates organic aerosol in CESM2.1 over the United States by comparing the model to long term surface measurements and aircraft campaigns. The authors find that the model overestimate organic aerosol during summer. Moreover, the model comparison with flight campaigns reveal that the model underestimate organic aerosol in the upper air. The authors conclude that these results could be explained by too high monoterpene SOA yields which result in too strong SOA production close to monoterpene sources. The topic of this paper falls well in the scope of ACP. The scientific methods in the paper are sound and well explained. The authors explain and discuss the results in the figures and tables in a clear and interesting manner. I recommend the paper for publication in ACP after the following comments have been addressed.

**General Response:** We greatly appreciate the referee for his/her time and efforts devoted to the review of our submission. We realize that most of the comments are due to the missing details of model description. We will present these details in this document as shown in the following responses.

**Specific comments and responses:**

**Comment#1:** The model used in this study, CAM6-Chem differs from the standard CAM6 since it has a more advanced chemistry. My impression is that CAM6 is the standard atmospheric model in CESM2.1. Could you describe to what extent CAM6-Chem is used in comparison to CAM6? It would be beneficial to better clarify the differences between CAM6 and CAM6-Chem in the methods section. In part of the method, you describe the changes in CESM2.1. It would be nice to refer to CAM6 or CAM6-Chem instead, as the text is currently written it is difficult to know if the VBS scheme is included in both CAM6 and CAM6-Chem or only in the latter. Moreover, you have evaluated CAM6-Chem, but it would be interesting to know how well CAM6 performs in comparison to CAM6-Chem with respect to organic aerosol. What are the differences in performance between CAM6-Chem and CAM6 in terms of organic aerosol?

**Response:** CESM2 (versions 2.0 and 2.1) supports two atmospheric model configurations, the Whole Atmosphere Community Climate Model version 6 (WACCM6) with 72 vertical layers up to about 150 km and the Community Atmosphere Model version 6 (CAM6) with 32 vertical layers up to about 40 km. CAM6 has simplified chemistry and simplified OA scheme, while CAM6 with comprehensive chemistry and comprehensive OA scheme are called CAM6-Chem. The differences between CAM6 and CAM6-Chem are included in the methods section. Due to the simplified chemistry and simple OA scheme, CAM has been used to explore physical processes, like cloud and precipitation processes (English et al., 2014), while CAM-Chem has been used to simulate specific chemical species and explore its climate effect (Schwantes et al., 2020; Tilmes et al., 2019; Jo et al., 2020). Our study focuses on the simulation performance of OA presented with VBS scheme, which is only included in CAM-Chem, not in CAM. We have changed the term "CESM2.1" to "CAM6-Chem" in method section and latter parts to avoid misunderstanding according to this comment. More detailed description of the differences between CAM and CAM-chem are reported in Tilmes et al. (2019), which comprehensively compared the difference between CAM6, WACCM, and WACCM6-Chem. WACCM6-Chem has almost exactly the same chemistry with very minor difference. Regarding the different performance of organic aerosol between CAM6-chem and CAM: CAM uses prescribed aerosol without comprehensive chemistry or SOA scheme and the simulation of aerosol is only used to serve radiative forcing and cloud, thus CAM is not compared to observation for organic aerosol.

**Reference**

English, J., Kay, J., Gettelman, A., Liu, X., Wang, Y., Zhang, Y., and Chepfer, H.: Contributions of Clouds, Surface Albedos, and Mixed-Phase Ice Nucleation Schemes to Arctic Radiation Biases in CAM5, Journal of Climate, 27, 5174–5197, 10.1175/JCLI-D-13-00608.1, 2014.

Jo, D. S., Hodzic, A., Emmons, L. K., Tilmes, S., Schwantes, R. H., Mills, M. J., Campuzano-Jost, P., Hu, W., Zaveri, R. A., Easter, R. C., Singh, B., Lu, Z., Schulz, C., Schneider, J., Shilling, J. E., Wisthaler, A., and Jimenez, J. L.: Future changes in isoprene-epoxydiol-derived secondary organic aerosol (IEPOX-SOA) under the shared socioeconomic pathways: the importance of explicit chemistry, Atmos. Chem. Phys. Discuss., 2020, 1-57, 10.5194/acp-2020-543, 2020.

Schwantes, R. H., Emmons, L. K., Orlando, J. J., Barth, M. C., Tyndall, G. S., Hall, S. R., Ullmann, K., St. Clair, J. M., Blake, D. R., Wisthaler, A., and Bui, T. P. V.: Comprehensive isoprene and terpene gas-phase chemistry improves simulated surface ozone in the southeastern US, Atmospheric Chemistry and Physics, 20, 3739-3776, 10.5194/acp-20-3739-2020, 2020.

Tilmes, S., Hodzic, A., Emmons, L. K., Mills, M. J., Gettelman, A., Kinnison, D. E., Park, M., Lamarque, J. F., Vitt, F., Shrivastava, M., Campuzano-Jost, P., Jimenez, J. L., and Liu, X.: Climate Forcing and Trends of Organic Aerosols in the Community Earth System Model (CESM2), Journal of Advances in Modeling Earth Systems, 10.1029/2019ms001827, 2019.

**Conment#2**: The model is only compared to observations over the United States. Could you comment on the limitations of this and if the model has been evaluated in other locations in any other studies.

**Response:** In this manuscript we focused on evaluation over the United States mainly because it has the best public accessible long-term observation data to support the evaluation. The aim of this paper is to reveal the performance of CESM2.1 in OA simulation which has not been thoroughly discussed in other studies to the best of our knowledge. Tsigaridis et al. (2014) and Tilmes et al. (2019) validated the model against flight campaign data over North America, the Pacific, and Atlantic. Dong et al. (2018) validated the performance of CAM-chem over Europe and Asia for $O_3$, $PM_{2.5}$, $PM_{10}$, and aerosol optical depth (AOD) and reported the model showed comparable performance with other popular global models such as EMEP and GEOS5. Gaubert et al. (2020) validated the model performance for CO over South Korea against the KORUS-AQ flight campaign data. Similar to other global models, CAM-chem has been widely applied in global-scale studies thus it is usually validated against flight campaign data or satellite products (Gliß et al., 2021; Kim et al., 2019) other than with regional scale measurements. Pfister et al. (2020) indicated a regional refine version of CAM-chem will be applied for regional scale air quality studies, and we are expecting to perform such type of simulation and probe into the model performance at regional scale over East Asia.

**Reference:**

Dong, X., Fu, J. S., Zhu, Q., Sun, J., Tan, J., Keating, T., Sekiya, T., Sudo, K., Emmons, L., Tilmes, S., Jonson, J. E., Schulz, M., Bian, H., Chin, M., Davila, Y., Henze, D., Takemura, T., Benedictow, A. M. K., and Huang, K.: Long-range transport impacts on surface aerosol concentrations and the contributions to haze events in China: an HTAP2 multi-model study, Atmos. Chem. Phys., 18, 15581–15600, https://doi.org/10.5194/acp-18-15581-2018, 2018.

Gliß, J., Mortier, A., Schulz, M., Andrews, E., Balkanski, Y., Bauer, S. E., Benedictow, A. M. K., Bian, H., Checa-Garcia, R., Chin, M., Ginoux, P., Griesfeller, J. J., Heckel, A., Kipling, Z., Kirkevåg, A., Kokkola, H., Laj, P., Le Sager, P., Lund, M. T., Lund Myhre, C., Matsui, H., Myhre, G., Neubauer, D., van Noije, T., North, P., Olivié, D. J. L., Rémy, S., Sogacheva, L., Takemura, T., Tsigaridis, K., and Tsyro, S. G.: AeroCom phase III multi-model evaluation of the aerosol life cycle and optical properties using ground- and space-based remote sensing as well as surface in situ observations, Atmos. Chem. Phys., 21, 87–128, https://doi.org/10.5194/acp-21-87-2021, 2021.

Kim, D., Chin, M., Yu, H., Pan, X., Bian, H., and Tan, Q.: Asian and trans-pacific dust: A multimodel and multiremote sensing observation analysis, J. Geophys. Res.-Atmos, 124, 13534–13559, https://doi.org/10.1029/2019JD030822, 2019

Pfister, G. G., Eastham, S. D., Arellano, A. F., Aumont, B., Barsanti, K. C., Barth, M. C., Conley, A., Davis, N. A., Emmons, L. K., Fast, J. D., Fiore, A. M., Gaubert, B., Goldhaber, S., Granier, C., Grell, G. A., Guevara, M., Henze, D. K., Hodzic, A., Liu, X., Marsh, D. R., Orlando, J. J., Plane, J. M. C., Polvani, L. M., Rosenlof, K. H., Steiner, A. L., Jacob, D. J., and Brasseur, G. P.: The Multi-Scale Infrastructure for Chemistry and Aerosols (MUSICA), Bulletin of the American Meteorological Society, 101, E1743-E1760, 10.1175/BAMS-D-19-0331.1, 2020.

Tilmes, S., Hodzic, A., Emmons, L. K., Mills, M. J., Gettelman, A., Kinnison, D. E., Park, M., Lamarque, J. F., Vitt, F., Shrivastava, M., Campuzano-Jost, P., Jimenez, J. L., and Liu, X.: Climate Forcing and Trends of Organic Aerosols in the Community Earth System Model (CESM2), Journal of Advances in Modeling Earth Systems, 10.1029/2019ms001827, 2019.

Tsigaridis, K., Daskalakis, N., Kanakidou, M., Adams, P. J., Artaxo, P., Bahadur, R., Balkanski, Y., Bauer, S. E., Bellouin, N., Benedetti, A., Bergman, T., Berntsen, T. K., Beukes, J. P., Bian, H., Carslaw, K. S., Chin, M., Curci, G., Diehl, T., Easter, R. C., Ghan, S. J., Gong, S. L., Hodzic, A., Hoyle, C. R., Iversen, T., Jathar, S., Jimenez, J. L., Kaiser, J. W., Kirkevåg, A., Koch, D., Kokkola, H., Lee, Y. H., Lin, G., Liu, X., Luo, G., Ma, X., Mann, G. W., Mihalopoulos, N., Morcrette, J.-J., Müller, J.-F., Myhre, G., Myriokefalitakis, S., Ng, N. L., O'Donnell, D., Penner, J. E., Pozzoli, L., Pringle, K. J., Russell, L. M., Schulz, M., Sciare, J., Seland, Ø., Shindell, D. T., Sillman, S., Skeie, R. B., Spracklen, D., Stavrakou,

T., Steenrod, S. D., Takemura, T., Tiitta, P., Tilmes, S., Tost, H., van Noije, T., van Zyl, P. G., von Salzen, K., Yu, F., Wang, Z., Wang, Z., Zaveri, R. A., Zhang, H., Zhang, K., Zhang, Q., and Zhang, X.: The AeroCom evaluation and intercomparison of organic aerosol in global models, Atmos. Chem. Phys., 14, 10845–10895, https://doi.org/10.5194/acp-14-10845-2014, 2014.

**Technical corrections and responses:**

**Comment#3:** Line 71:"over previous versions" sounds a bit odd.

**response:** We have revised the sentence at line 71 as shown below.

CESM2 (versions 2.0 and 2.1) includes 2 versions of model top, the Whole Atmosphere Community Climate Model version 6 (WACCM6) with 72 vertical layers up to about 150 km and the Community Atmosphere Model version 6 (CAM6) with 32 vertical layers up to about 40 km. CAM6 has simplified chemistry and simplified OA scheme, while CAM6 with comprehensive chemistry and comprehensive OA scheme are called CAM6-Chem which is updated compared to previous versions.

**Comment#4:** Line 138-141: This is a very long sentence, please split it up.

**Response:** We have split the sentence at line 139-141 as shown below.

To exclude the influence of potential extreme meteorology condition or emission inputs, these sensitivity runs are configured with FC2010climo component set and Newtonian relaxation time of three hours. The FC2010climo component set is as same as FCSD component set except that the emissions are a 10-year average used for each year of the simulation.

**Comment#5:** Line 169: "prominently overestimates in" is there a word missing here? What is overestimated?

**Response:** We apologize for the missing word in the sentence at line 169. We have corrected the sentence as shown below.

In EUS, the simulation prominently overestimates surface OA concentration in summer by 4.26 μg/m$^3$ (131.15 %) but successfully reproduces the temporal change with a strong correlation with observations of 0.60 (Table 3) as shown in Fig. 2(c).

**Comment#6:** Line 170: "with a strong correlation with observations of 0.60 as shown in Fig. 2c" The correlation coefficients are not shown in Fig 2 but rather in Table 3. Please refer to the table or both the figure and table. The same problem with referring to figure 2 instead of table 3 occur on line 174.

**Response:** We apologize for the problem at line 170 and line 174. Please see the response of comment#5 for the modification of the sentence at line 170. The modification of the sentence at line 174 is shown below.

The model shows smaller bias in WUS but also a poor correlation of 0.36 (Table 3) in summer as shown in Fig. 2(e) and 2(f).

**Comment#7:** Line 170: "As compared with CONUS domain, simulation at" are there missing words in this part of the sentence?

**Response:** We apologize for the missing words in the sentence at line 170. The "simulation" is referred to the "surface OA concentration from the simulation". We have modified the sentence as follows.

As compared with CONUS domain, surface OA concentration from the simulation at EUS shows an even greater overestimation during warmer months as shown in Fig. 2(d).

**Comment#8:** Line 295-299. This is a very long sentence that should probably be split up. Also, the English in this sentence needs to be checked.

**Response:** We totally agree that the long sentence is poorly readable. We have split and modified the long sentence in line 295-299 as follows.

It is certainly reasonable to take the wall-loss effect into account when making the chamber measurements. But it also should be noticed that those measurements were conducted under artificial environment with predefined chemical species that may vary significantly from the real meteorology condition and atmospheric chemistry regime. Thus, the parameters reported in the chamber studies need to be carefully interpreted and adjusted when applied in atmospheric models.

**Reply for the referee comment#2**

**General comments and responses:**

**Comment#1.** The argument that the model overestimation of OA at the surface and in summer is caused by an overrepresentation of monoterpene SOA production needs to be better substantiated or the limitations of this assertion better discussed. To be clear, I think the authors make a good suggestion by pointing to the large monoterpene SOA burden and the correlation with model bias. However, the correlation with model bias is not enough to make this argument. For instance, the authors point out that isoprene SOA also has a positive correlation with model bias. I don't completely follow why isoprene SOA was dismissed as a reason (Line 365). Isoprene also appears to have a seasonal cycle that peaks in the summer and the authors note the correlation with isoprene and model bias is also positive. Could the over prediction be due to both monoterpene and isoprene SOA yields? Additionally, the authors state in the Conclusions (Line 362) that the other POA and SOA components cannot explain the model bias; however, this does not appear to be explicitly shown in the Results section. I do see that monoterpene SOA dominants the OA composition in the summer (and I agree this is a good candidate for the cause of model overestimation), but I do not see a discussion that the other SOA species could not also contribute to the model overestimation. Again to be clear, I agree that the monoterpene SOA yield is a good suggestion for the cause of the model overestimation, but I feel this argument needs more context. This paper could be improved with a better discussion of this argument and its limitations or an additional simulation with altered monoterpene yields that reduced the model bias.

**Response:** We thank the reviewer for pointing out the limitation of the assertion and we agree that more discussion is necessary to clearly explain and describe the result. Our conclusion is that monoterpene-derived SOA is the largest contributor to modeling bias, thus it is urgently needed to improve the related chemical mechanism in CAM-chem. This conclusion was derived through two steps: first, we found that the spatiotemporal characteristics of simulation bias (Sect. 3.1: the eastern U.S. showed larger bias than the western U.S., summer showed larger bias than winter) indicated the biogenic VOCs played a dominant role over anthropogenic VOCs, because anthropogenic emissions of VOCs and POA wouldn't lead to such spatiotemporal characteristics. Second, the first finding inspired us to conduct sensitivity simulations to quantitatively estimate the contributions from every precursor through each chemical pathway to SOA (Sect. 3.2), and we found the monoterpene-derived SOA consisted 43% of the total SOA in summer, while none of the other precursors contributed more than 20%. Consequently, the 68% overestimation in summer shall most likely be attributed to monoterpene-related uncertainty. This comment reminds us that monoterpene might be a good candidate but other possibilities cannot be

completely excluded. Just like the referee pointed out, the secondary largest contributor is isoprene-derived SOA, consist 17.0 % of total SOA in summer. Thus theoretically, the uncertainty in isoprene-derived SOA can also play an important role in the modeling bias. According to this comment, we have added a few brief discussions and rephrased our assertions in the manuscript. We also added discussion about the potential uncertainties related with isoprene-SOA treatment at line 329-340 with Figure 7. We have revised the paragraph at line 360-368 in the Sect. 4 as follows.

"Our analysis suggests that it is likely that simulated monoterpene SOA production is parameterized with too high yields and may be the most influential factor that affects the modeling bias for three reasons: first, monoterpene SOA contributes most (46.3 %) to the total SOA in summertime, while other anthropogenic POA or BVOCs have substantially smaller contributions. The large contribution of monoterpene SOA simulated by CAM6-Chem is consistent with other measurement and modeling studies, but the current VBS configuration adopted from GEOS-Chem may require further adjustment. Isoprene may also play an important role in modeling uncertainty but the influence is likely less significant than monoterpene as the isoprene-derived SOA consists 17.0 % of total SOA in summer. Second, the simulation bias showed a strong spatiotemporal correlation with monoterpene emission as demonstrated by the large overestimation in summer over eastern US, and larger overestimation of OA is found at places with higher $NO_x$ condition under same monoterpene emissions level. Third, overestimation of OA at surface layer and underestimation of OA and monoterpene in the free troposphere suggests that both the production and photolytic removal processes might be parameterized too strong."

**Comment#2.** A large limitation of this study is that it focuses only on SOA production as opposed to any other chemical or physical processes in the model. Could the over estimation of model OA at the surface and underrepresentation of OA aloft point to an issue with vertical transport or removal? What about evaporation of OA - is this included in the model? I believe this paper would benefit from an explicit discussion of this limitation and how it affects the results.

**Response:** We agree with the reviewer the simulated mass concentration of OA is affected by many processes in addition to the chemical production of SOA. Other processes may also affect SOA, such as the biogenic and anthropogenic emissions, photolytic removal, dry and wet depositions, mixing, and transport. Modeling treatment for any of these processes may induce uncertainty to the final simulated mass concentrations of SOA. Lifetime of OA is about less than one week (Tsigaridis et al., 2014) owing to the rapid reaction rates of VOCs and the oxidants (OH, $O_3$, $NO_3$), thus the chemical production may play a more important role than other processes in the model. We intended to explore the remaining

uncertainties within the chemical production because research community has been focusing on this topic during the past decades due to continuously improved understandings and new findings from lab and chamber studies. Development of SOA chemical production treatment in CAM-chem has been discussed in Tilmes et al. (2019) and Emmons et al. (2020), and there is also another paper by Jo et al. (2021) which reported the most recent update of SOA chemistry in CESM. But none of these studies provide a thorough evaluation of the model against surface observations at long-term scale, and a thorough understanding of the modeling bias is in absence. This is the major reason motivating our study, because it will help to clearly understand the performance and identify the most important causes for remaining uncertainties with the SOA chemistry. In contrast, other processes such as the deposition and mixing treatment of CESM are relatively better understood based on previous studies (Tilmes et al., 2019; Tsigaridis et al., 2014). It is inapplicable to elucidate all the related emission, physical, and chemical processes to identify and evaluate every potential source of uncertainties in one publication because that would be too long. Also, currently there is no exactly right answer for the SOA related processes as our understanding is still under development (Shrivastava et al., 2017). Thus, whenever the model could be adjusted to better describe the observed concentration of SOA, we have to make sure it is do so for the right reason. Identify and narrow down the bias in SOA chemistry can help us to build a better baseline, to further evaluate the uncertainties from other factors one by one and step by step.

We agree with the reviewer that vertical transport may affect the simulation results of VOCs and SOA. CESM can generally well reproduce the vertical profiles of meteorology variables (He et al., 2015). A recent sensitivity study (Gaubert et al., 2020) demonstrated the systematic underestimation of CO vertical profile in Asia was mainly due to bias in anthropogenic emission instead of the vertical transport scheme. In our case, the model showed larger upper air underestimations in the Eastern U.S. (EUS) than the Western U.S. (WUS), and likewise for surface overestimations. If vertical transport played an important role, it would more likely induce systematic bias leading to similar scale of errors in EUS and WUS. Thus, the discrepancy in vertical profile is more likely related to VOCs chemistry rather than transport issue.

Evaporation of OA is represented in the CESM model. One of the main functions of VBS is to calculate the thermodynamic equilibrium between gas-phase precursors (SOAG) and SOA, based on the volatilities of different species which are categorized into 5 groups (details were listed in Table 4) in CESM. Evaporation is not treated for primary emissions of OA (POA) in current version of CESM.

**Reference:**

Emmons, L. K., Schwantes, R. H., Orlando, J. J., Tyndall, G., Kinnison, D., Lamarque, J. F., Marsh, D., Mills, M. J., Tilmes, S., Bardeen, C., Buchholz, R. R., Conley, A., Gettelman, A., Garcia, R., Simpson,

I., Blake, D. R., Meinardi, S., and Pétron, G.: The Chemistry Mechanism in the Community Earth System Model Version 2 (CESM2), Journal of Advances in Modeling Earth Systems, 12, 10.1029/2019ms001882, 2020.

Gaubert, B., Emmons, L. K., Raeder, K., Tilmes, S., Miyazaki, K., Arellano, A. F., Jr., Elguindi, N., Granier, C., Tang, W., Barre, J., Worden, H. M., Buchholz, R. R., Edwards, D. P., Franke, P., Anderson, J. L., Saunois, M., Schroeder, J., Woo, J. H., Simpson, I. J., Blake, D. R., Meinardi, S., Wennberg, P. O., Crounse, J., Teng, A., Kim, M., Dickerson, R. R., He, H., Ren, X., Pusede, S. E., and Diskin, G. S.: Correcting model biases of CO in East Asia: impact on oxidant distributions during KORUS-AQ, Atmos Chem Phys, 20, 14617-14647, 10.5194/acp-20-14617-2020, 2020.

He, J., Zhang, Y., Glotfelty, T., He, R., Bennartz, R., Rausch, J., and Sartelet, K.: Decadal simulation and comprehensive evaluation of CESM/CAM5.1 with advanced chemistry, aerosol microphysics, and aerosol-cloud interactions, Journal of Advances in Modeling Earth Systems, 7, 110-141, 10.1002/2014ms000360, 2015.

Jo, D. S., Hodzic, A., Emmons, L. K., Tilmes, S., Schwantes, R. H., Mills, M. J., Campuzano-Jost, P., Hu, W., Zaveri, R. A., Easter, R. C., Singh, B., Lu, Z., Schulz, C., Schneider, J., Shilling, J. E., Wisthaler, A., and Jimenez, J. L.: Future changes in isoprene-epoxydiol-derived secondary organic aerosol (IEPOX SOA) under the Shared Socioeconomic Pathways: the importance of physicochemical dependency, Atmos. Chem. Phys., 21, 3395-3425, 10.5194/acp-21-3395-2021, 2021.

Shrivastava, M., Cappa, C. D., Fan, J., Goldstein, A. H., Guenther, A. B., Jimenez, J. L., Kuang, C., Laskin, A., Martin, S. T., Ng, N. L., Petaja, T., Pierce, J. R., Rasch, P. J., Roldin, P., Seinfeld, J. H., Shilling, J., Smith, J. N., Thornton, J. A., Volkamer, R., Wang, J., Worsnop, D. R., Zaveri, R. A., Zelenyuk, A., and Zhang, Q.: Recent advances in understanding secondary organic aerosol: Implications for global climate forcing, Reviews of Geophysics, 55, 509-559, 10.1002/2016rg000540, 2017.

Tilmes, S., Hodzic, A., Emmons, L. K., Mills, M. J., Gettelman, A., Kinnison, D. E., Park, M., Lamarque, J. F., Vitt, F., Shrivastava, M., Campuzano-Jost, P., Jimenez, J. L., and Liu, X.: Climate Forcing and Trends of Organic Aerosols in the Community Earth System Model (CESM2), Journal of Advances in Modeling Earth Systems, 10.1029/2019ms001827, 2019.

Tsigaridis, K., Daskalakis, N., Kanakidou, M., Adams, P. J., Artaxo, P., Bahadur, R., Balkanski, Y., Bauer, S. E., Bellouin, N., Benedetti, A., Bergman, T., Berntsen, T. K., Beukes, J. P., Bian, H., Carslaw, K. S., Chin, M., Curci, G., Diehl, T., Easter, R. C., Ghan, S. J., Gong, S. L., Hodzic, A., Hoyle, C. R., Iversen, T., Jathar, S., Jimenez, J. L., Kaiser, J. W., Kirkevåg, A., Koch, D., Kokkola, H., Lee, Y. H., Lin, G., Liu, X., Luo, G., Ma, X., Mann, G. W., Mihalopoulos, N., Morcrette, J. J., Müller, J. F., Myhre, G.,

Myriokefalitakis, S., Ng, N. L., amp, apos, Donnell, D., Penner, J. E., Pozzoli, L., Pringle, K. J., Russell, L. M., Schulz, M., Sciare, J., Seland, Ø., Shindell, D. T., Sillman, S., Skeie, R. B., Spracklen, D., Stavrakou, T., Steenrod, S. D., Takemura, T., Tiitta, P., Tilmes, S., Tost, H., van Noije, T., van Zyl, P. G., von Salzen, K., Yu, F., Wang, Z., Wang, Z., Zaveri, R. A., Zhang, H., Zhang, K., Zhang, Q., and Zhang, X.: The AeroCom evaluation and intercomparison of organic aerosol in global models, Atmospheric Chemistry and Physics, 14, 10845-10895, 10.5194/acp-14-10845-2014, 2014.

**Comment#3.** The sensitivity simulations turn off chemical reactions one at a time; however, these chemical mechanisms are not necessarily linear (or additive). How does this assumption affect your results?

**Response:** We agree with the reviewer that many of the reactions are nonlinear, directly removing one reaction may drive the full mechanism to a different chemical regium and induce uncertainty to our estimation. To avoid the nonlinear-related uncertainty, in each of the sensitivity simulations we turned off a specific reaction by artificially set the SOA precursors (SOAG) yield to zero. This means the reaction was still simulated together with all other reactions, but the related SOAG was not produced. The related reactants (mainly OH, O3, and NO3) remains unaffected. For example, the reaction of monoterpene (MTERP) oxidized by OH is represented as:

MTERP + OH -> MTERP + OH + 0.0508*SOAG0 + 0.1149*SOAG1 + 0.0348*SOAG2 + 0.0554*SOAG3 + 0.1278*SOAG4

In the sensitivity simulation, the reaction was represented as:

MTERP + OH -> MTERP + OH + 0.0000*SOAG0 + 0.0000*SOAG1 + 0.0000*SOAG2 + 0.0000*SOAG3 + 0.0000*SOAG4

Gas phase MTERP and OH were still cycled through the sensitivity simulation, and since the precursors (SOAG) didn't affect each other or get involved in chemical transformation, nonlinear uncertainty was not induced to the full chemical mechanism in each sensitivity simulation.

**Comment#4.** The Introduction Section would benefit from further discussion on SOA oxidation and chemistry (including the VBS scheme) as well as a literature review of previous studies focused on model/measurement comparison of OA. Additionally, a number of statements are lacking citations (see Specific Comments for examples). This topic (of OA representation in models) has been explored

previously. As I mentioned at the start of this review, the continued challenge of representing OA in models certainly warrants continued study. That said, I believe this study could be improved by including a literature review of previous measurement/modeling studies in the Introduction. I note that the authors do point to and comment on previous studies in the Methods and Results Sections (which is great). However, I think the manuscript would be improved by clearly discussing relevant previous work in the introduction. This would improve the ability of readers to follow the comparisons in later sections. One such example of this, is that Hodzic et al. (2016), which is cited throughout this work, seems to argue for stronger SOA production rates and stronger SOA sinks. Conversely, this study seems to argue for the opposite. This is an interesting comparison that could use more context.

**Response:** We agree with the reviewer that a clearly discussion of relevant previous work in the introduction section would be helpful to present a big picture of the whole SOA modeling study. The first version of CAM-chem was introduced by Lamarque et al. (2012) in which the SOA chemistry was represented with the two-product method (Lack et al., 2004; Heald et al., 2008). The next big update was reported by Tilmes et al. (2019), in which the two-product method was replaced by VBS following the work by Hodzic et al. (2016). Although CAM-chem has been applied in many studies including the AeroCom program (Tsigaridis et al., 2014), the model hasn't been thoroughly evaluated against long-term ground-based measurements. Tilmes et al. (2019) validated the model with flight campaign data, which was a very commonly used and well-accepted method for global model evaluation. This method gives a snapshot of the mode performance since campaigns are limited in time period and space coverages. As the global models are getting finer grid resolution and more detailed description of atmospheric chemistry processes, it is necessary to reveal the performance with more details to help understand the remaining uncertainty after lots of new model developments. We have added this in the revised manuscript.

We greatly appreciate the time and efforts the referee devoted to review our submission. Hodzic et al. (2016) indeed argued for stronger SOA production and stronger sink through photolytic depletion, but that VBS implementation was based on GEOS-Chem. Since the gas-phase chemical mechanisms, dry and wet deposition schemes, heterogenous chemistry schemes are all different between GEOS-Chem and CAM-chem, even the same configuration of VBS may lead to different simulation results from two distinct modeling platforms. So, the stronger production configuration demonstrated in Hodzic et al. (2016) may not necessarily lead to good performance in CAM-chem, as has been revealed through our study in line 293-304. To identify the exact causes for different VBS performance the two models would require a comprehensive comparison between them. We are planning to do such a comparison in the future, but this manuscript is trying to understand the uncertainties in CAM-chem, and an explicit comparison is out of the scope of this study. We agree with the reviewer that a brief comparison and

explanation is necessary in the context, and we have added such discussions in the introduction section.

Please see the modification of the paragraph at line 49-57 as follows.

"Modeling discrepancies largely come from the lack of a consensus in the representation of chemical composition and formation processes of OA among different models (Tsigaridis et al., 2014; Goldstein and Galbally, 2007). Although laboratory and chamber studies have explicitly revealed thousands of new reactions and new species related to VOCs and SOA, these reactions and species are usually simplified and grouped into a few functions and lumped to fewer species in the models to make it possible for numeric simulating. In addition, many unclear SOA formation processes have to be approximated as the knowledge is still under development (Kanakidou et al., 2004; Hallquist et al., 2009). Thus, different models may use different simplified functions, definitions for lumped species, and approximation methods to represent the same SOA related processes. SOA production 
[revised manuscript text omitted]

**Specific comments and responses:**

**Comment#1.** I do not think an appropriate color scale was chosen for Figure 1. While red and blue are certainly appropriate for opposite ends of a diverging color scale, it is not immediately obvious to me that green should be opposite yellow. I suggest the authors use a standard diverging color scale (or simply shades of red and blue on each side).

**Response:** We intended to use warm colors for positive bias and cool colors for negative bias. We have changed the figure to use red and blue as shown below.

**Figure 1: 1998–2019 CAM-Chem-SD surface OA concentration mean bias (top figure, unit: μg/m³) and normalized mean bias (bottom figure, unit: %) compared to IMPROVE data. CONUS is divided into two subdomains, EUS (red box) and WUS (blue box).**

**Comment#2.** The units of the color scale in Figure 1 could be more obvious. I suggest including the units as the color bar label itself or at least in the caption.

**Response:** We apologize for improper color label and missing units in Fig. 1. We modify color labels and include the units of each subplot in the Fig. 1 and its caption. The caption is revised as shown in the response for specific comment#1.

**Comment#3.** The legend of Figure 4 is much too small to be readable.

**Response:** We apologize for the unclear legends in Fig.4. We now use bigger size of legend and higher DPI for the whole figure, thus readers can zoom in it if necessary. Please see the figure below.

[Figure]

**Figure 4: (a)~(c) Seasonal variation of surface OA concentration from CAM-Chem-climo simulation at all IMPROVE sites located over (a) CONUS, (b) EUS and (c) WUS. The contribution of POA and SOA in five VBS bins are shown in filled areas. The 14-month average contribution of POA and SOA are shown as the number in the legend. (d)~(f) Seasonal variation of SOA formed by 15 different pathways over (d) CONUS, (e) EUS and (f) WUS. The contribution of 15 reactions are shown in filled areas and 14-month average contribution is shown as**

the number in the legend. (g)~(i) the contributions from each pathway to the five SOA bins over (g) CONUS, (h) EUS and (i) WUS in 2010 July. SOA concentration in 5 bins are shown in filled bars.

**Comment#4.** The panel labels in the caption of Figure 4 are inconsistent. Sometimes the panel label follows the description while other times it precedes the description.

Response: We apologize for the inconsistent panel label of Fig. 4. We modify the description of Fig.4 followed by the panel label. Please see the response of comment#3.

**Comment#5**. The caption in Figure 7 points to the wrong color (I think it should read "black line" instead of red).

Response: Thanks for your comments. We modify the caption of Fig. 7. Please see the figure and its caption below.

[Figure]

**Figure 7: 1988–2019 JJA (a) simulated surface OA decade trend and IMPROVE surface OA decade trend at selected IMPROVE sites (filled circles). Black circles indicate sites with statistically significant trends with 95 % confidence according to the student's T-test; (b) simulated (black lines) and observed (red dots) surface OA concentration at Cape Romain NWR site (shown as purple circle in Fig. 7(a)). Simulated and observed OA decade trends (unit: μg/m³/10 year) in summer at the site are shown as the numbers in the legend.**

**Comment#6.** The units in Figure 7a are never stated. Please be explicit about units in all figures.

**Response:** We apologize for the typo in the figures. We include units in Fig. 7 and its caption. Please see the response of comment#5.

**Comment#7:** Table 3 is difficult to read. Could the different regions be grouped in a more obvious way?

**Response:** We modify Table 3 by adding extra space between different regions, as shown below.

Table 3: The correlation coefficient (CC), mean bias (MB) and normalized mean bias (NMB) between observations (five field campaigns and IMPROVE surface measurements) and CAM-Chem-SD.

| Observations | | CC | Mean Obs. ($\mu g/m^3$) | Mean Sim. ($\mu g/m^3$) | MB ($\mu g/m^3$) | NMB (%) |
|---|---|---|---|---|---|---|
| | | | IMPROVE | | | |
| CONUS | Annual | 0.40 | 2.07 | 2.48 | 0.41 | 20.27 |
| | Spring | 0.67 | 1.65 | 1.57 | -0.08 | -4.81 |
| | Summer | 0.37 | 2.90 | 4.87 | 1.97 | 68.78 |
| | Fall | 0.34 | 2.23 | 2.23 | -0.01 | 0.13 |
| | Winter | 0.70 | 1.49 | 1.22 | -0.27 | -19.05 |
| | | | | | | |
| EUS | Annual | 0.79 | 2.64 | 3.72 | 1.08 | 40.82 |
| | Spring | 0.64 | 2.49 | 2.66 | 0.17 | 6.71 |
| | Summer | 0.60 | 3.26 | 7.52 | 4.26 | 131.15 |
| | Fall | 0.69 | 2.63 | 2.88 | 0.25 | 9.70 |
| | Winter | 0.82 | 2.25 | 1.82 | -0.43 | -19.11 |
| | | | | | | |
| WUS | Annual | 0.36 | 1.78 | 1.89 | 0.11 | 10.49 |
| | Spring | 0.77 | 1.23 | 1.04 | -0.19 | -15.36 |
| | Summer | 0.48 | 2.72 | 3.66 | 0.94 | 34.83 |
| | Fall | 0.35 | 2.03 | 1.91 | -0.12 | -5.85 |
| | Winter | 0.73 | 1.12 | 0.91 | -0.21 | -18.00 |
| | | | Aircraft | | | |

| | | | | | |
|---|---|---|---|---|---|
| CalNex | 0.43 | 2.06 | 1.46 | -0.60 | -29.01 |
| DC3 | 0.12 | 2.99 | 0.72 | -2.17 | -72.75 |
| SENEX | 0.33 | 7.09 | 4.22 | -2.87 | -40.54 |
| SEAC4RS | 0.10 | 6.90 | 1.93 | -4.97 | -71.97 |
| FRAPPE | 0.27 | 3.05 | 2.42 | -0.63 | -20.64 |

**Comment#8.** Please add units to Table 3.

**Response:** We add the units into Table 3 as shown in the response of comment#7.

**Comment#9.** This is a minor comment - the citations need spaces after the semicolon.

**Response:** Thanks for this comment. We add spaces after the semicolon in the citations.

**Comment#10.** The sentence at Line 51 is unclear. Are you comparing the model representation to other model processes or stating the reasons why model representation of OA is challenging?

**Response:** We apologize for the unclear description. In this sentence, we intended to state that model discrepancies are due to different representations of OA among the models, and this is because our knowledge of VOCs and SOA related chemistry is still under development and lacks a consensus understanding. We have revised the sentence as shown in the response for general comment#4.

**Comment#11.** Lines 51-57 should be edited or revised for clarity. I am not sure what is meant by these sentences.

**Response:** We totally agree with the unclear information at Line 51-57. In this paragraph, we want to clarify the model discrepancies comes from different SOA representations among models which are lack of some processes compared to chamber studies. We have revised the sentence as shown in the response for general comment#4.

**Comment#12**. Citations are needed for the comments on OA and climate impacts (such as on Line 41 and Line 60).

**Response:** We apologize for the missing citations for the comments on OA and climate impacts. Citations are included in the sentence at Line 41.

"The radiative forcing effect of OA has been assessed with climate models through tremendous efforts during the past decades (Ghan et al., 2012; Myhre et al., 2013; Sporre et al., 2020; Chen and Gettelman, 2016), yet the limited capability of climate models in terms of simulating the productions and depletions of OA induce large uncertainties."

Citations are also included in the sentence at Line 60.

"The model has been widely applied for OA climate effect assessment purpose (A. Gettelman et al., 2019; Jo et al., 2021; Glotfelty et al., 2017; Tilmes et al., 2019) and a significant portion of improvements have been implemented in the latest version regarding the chemical mechanisms (Tilmes et al., 2019), but evaluation of simulated OA concentration hasn't been well documented or thoroughly discussed."

**Comment#16**. Line 124 - Could you elaborate (briefly) on what a "specified dynamical" simulation is? In addition, please define "FCSD" and "FC2010climo".

**Response:** A "specified dynamical" simulation in Line 124 is referred to the FCSD simulation where meteorological fields is read from the input meteorological fields from MERRA2 reanalysis dataset for detailed comparisons to field experiments and specific observations. The meteorological fields here are referred to temperature, horizontal winds, and surface fluxes.

"FCSD" and "FC2000climo" are defined component sets of CESM which define meteorological field, emissions, physical and chemistry schemes used in each module. The information of "FCSD" and "FC2000climo" are shown in the table below. These 2 component sets differ in terms of emission and meteorological fields used in simulation.

| Component set | FCSD | FC2000climo |
|---|---|---|
| Emission | Historical emission depending on simulation period | Present day (year 2000) emission |
| Physics of atmosphere model | CAM6 | CAM6 |

| | CAM-Chem troposphere/stratosphere chemistry with simplified VBS-SOA | CAM-Chem troposphere/stratosphere chemistry with simplified VBS-SOA |
|---|---|---|
| Chemistry of atmosphere model | | |
| Meteorological fields | Nudged with MERRA2 dataset | Free running |
| Land model | CLM5 | CLM5 |

We have revised the sentence in line 125-127 as follows.

"This experiment (referred as CAM-Chem-SD) uses FCSD component set in which CAM6 physics, troposphere/stratosphere chemistry (MOZART-TS1) with VBS SOA scheme, historical emission, and offline meteorological field are applied. In details, Temperature, horizontal winds, and surface fluxes are nudged to Modern‐Era Retrospective analysis for Research and Applications (MERRA2) fields for detailed comparisons to field experiments and specific observations."

**Comment#17**. L 152 - I do not think the word choice of "critical" simulation bias is correct. Do you mean "large" or "substantial"?

**Response:** We agree with the uncorrected word used in the sentence at line 151-153. In the sentence at line 151-153, we are supposed to show apparent simulation bias of surface OA concentration as shown in Fig.1. We modify "critical simulation bias" to "substantial simulation bias" at line 152.

**Comment#18**. L 173 - I suggest changing "descending" trend to "decreasing" trend.

**Response:** We totally agree that the trend of simulated surface OA concentration in WUS is not obviously decreasing and modify "descending tread" to "decreasing trend" at line 173.

**Comment#19**. L 173 - I think this is a really interesting point that needs a little clarification. Do you mean that the increasing wildfires are leading to increasing trends in observations but the wildfire emissions are not included in the model and so the trend is not represented?

**Response:** Wildfire emissions are included in the model simulation, but the climate-fire-ecosystem interactions are not well represented in the climate models (Zou et al., 2020). For example, Zou et al. (2020) pointed out that wildfire is driven by offline statistical regression or one-way coupled prognostic fire models, while feedback to weather, climate, and vegetation was neglected in CESM.

In the sentence at line 173, we are supposed to show the opposite trend of surface OA concentration between simulation and IMPROVE dataset in summertime over WUS region. Wildfires lead to large inter-annual variation of observed surface OA concentration as discussed in Malm et al. (2017). The simulated OA also has large inter-annual variation but did not shows increasing trend due to lower value after 2017. The wildfire emissions from the CMIP6 emissions (Feng et al., 2020) are included in CAM-Chem-SD. It needs to be emphasized that historical emissions are used in CAM-Chem-SD simulation from 1987 to 2014 and SSP585 emissions after 2014 as clarify at line 129-132. The SSP585 emissions is based on shared socioeconomic pathway 5 (SSP5) (O'Neill et al., 2017) and forcing levels of Representative Concentration Pathways 8.5 (RCP8.5), which means the emissions do not exactly match the observed condition. Therefore, both of the simulated and observed OA show increasing trend in summertime over WUS region before 2014, but opposite trend after 2014. We have rephrased this sentence in 171-173 to avoid misunderstanding as follows.

"In WUS, simulated OA shows a slow decreasing trend in summer (-0.02 μg/m3 per decade) while the observations indicate an ascending trend (0.23 μg/m3 per decade). The large inter-annual variation from 1999 to 2019 are shown in observed surface OA concentration mainly due to the influence of wildfires (Malm et al., 2017). The simulated surface OA concentration also has large inter-annual variation but do not shows increasing trend due to lower value after 2017. It needs to be emphasized that historical emissions are used from 1987 to 2014 and SSP585 emissions after 2014 in CAM-Chem-SD simulation, which means the emissions do not exactly match the observed condition after 2014."

"Moreover, another simulation of CESM2.2 which is the latest released version of CESM, referred to CAM-Chem-SD (TS2), is conducted with MOZART-TS2 gas phase chemistry (Schwantes et al., 2020) from January 2013 to February 2014 with first 2 months as spin-up time. The FCSD component set is also used in CAM-Chem-SD (TS2). Except for the difference of gas phase chemistry, the SOA scheme is also improved in CAM-Chem-SD (TS2) compared with CAM-Chem-SD. The NOx dependence of SOA formation in CAM-Chem-SD (TS2) is not considered in CAM-Chem-SD. Thus, we compared CAM-Chem-SD and CAM-Chem-SD (TS2) to investigate the impact of NOx dependence on SOA formation."

We also modify Table 2 to include the experiment information of CAM-Chem-SD (TS2) as follows.

**Table 2: CESM experiments used in this study**

| Index | Experiment ID |
| --- | --- |
| B1 | CAM-Chem-SD |
| B2 | CAM-Chem-SD (TS2) |
| B3 | CAM-Chem-climo |
| E1 | no ISOP+OH |
| E2 | no ISOP+O3 |
| E3 | no ISOP+NO3 |
| E4 | no MTERP+OH |
| E5 | no MTERP+O3 |
| E6 | no MTERP+NO3 |
| E7 | no BCARY+OH |
| E8 | no BCARY+O3 |

| | |
|---|---|
| **E9** | no BCARY+NO3 |
| **E10** | no BENZENE+OH |
| **E11** | no TOLUENE+OH |
| **E12** | no XYLENES+OH |
| **E13** | no IVOC+OH |
| **E14** | no SVOC+OH |
| **E15** | no GLYOXAL |

In Sect. 3.3, we are discussing the possible factors that leads to the simulation bias of surface SOA concentration. The 2 major factors are strong monoterpene mass yields which is discussed before line 299 and NOx dependence in VBS scheme. CAM-Chem-SD(TS2) experiment is conducted to show that less SOA is formed when NOx dependence is considered in VBS scheme. Therefore, we compare CAM-Chem-SD and CAM-Chem-SD(TS2) in the paragraph begun at line 299 to discuss the influence of NOx dependence in SOA concentration. It shows when NOx dependence is considered in VBS scheme, surface OA concentration decrease but is still larger than observation, which means the strong monoterpene SOA yield is the main factor that leads to modeling bias.

**Comment#23**. Line 368 states "...and photolytic removal processes might be too strong". I do not follow why this is an argument in support of monoterpene SOA production being too high. It seems like it argues that bias is not entirely due to SOA production rates in contrast to the point of this paragraph.

**Response:** In this paragraph, we explained the three reasons of modeling bias in summertime. For the third reason, high SOA production rates lead to high SOA concentration, while strong photolytic removal processes lead to low SOA concentration. If both production and removal processes are configured too strong, the model is likely to show overestimation near source region (surface layer, near VOCs emission area) but underestimation in remote region (upper air, far from VOCs emission sources). Therefore, we conclude that CESM2.1 overestimated surface OA concentration and underestimated SOA concentration at higher altitude in summer.